# Score-Based Diffusion Modeling for Nonparametric Empirical Bayes in Heteroscedastic Gaussian Mixtures

**Gongyu Chen**
Department of IEOR
UC Berkeley
Berkeley, CA 94720
gongyu@berkeley.edu

**Ying Cui**
Department of IEOR
UC Berkeley
Berkeley, CA 94720
yingcui@berkeley.edu

## Abstract

We propose a generalized score-based diffusion framework for learning multivariate Gaussian mixture models with homoscedastic or heteroscedastic noise. Our goal is to nonparametrically estimate the latent location distribution and denoise the observations. Departing from the conventional maximum likelihood approach, we reinterpret each observation as a temporal slice of a family of stochastic diffusion processes. This modeling choice enables a principled characterization of the additive noise structure and supports a multi-step denoising procedure grounded in reverse-time dynamics. We introduce a score-based objective that explicitly models the latent distribution and accommodates observation-specific noise covariances. Theoretically, we establish that the score estimation error with $n$ independent observations achieves a near-parametric error rate of $\mathrm{polylog}(n)/n$, improving upon existing results in the diffusion literature. Empirically, our method outperforms the nonparametric maximum likelihood estimator in both density estimation and denoising fidelity, especially in high-dimensional settings. These findings suggest a promising direction for integrating nonparametric empirical Bayes with diffusion-based generative modeling for latent structure recovery.

## 1 Introduction

In many large-scale scientific domains, such as the analysis of high-dimensional astronomical data, measurements are often corrupted by *heteroscedastic* noise, where error variances vary across dimensions and differ across observations [Kelly, 2012, Akritas and Bershady, 1996, Anderson et al., 2018]. While this noise complicates downstream inference, it is frequently well-characterized through the data collection process. A central goal in such settings is to recover the true latent signals and their underlying distribution from noisy observations by leveraging structural information inherent in the data. This problem can be naturally modeled using heteroscedastic Gaussian location mixtures, described by:

$$X_i = \theta_i^* + Z_i, \qquad \text{with } \theta_i^* \overset{\text{i.i.d}}{\sim} G^* \text{ and } Z_i \overset{\text{ind}}{\sim} \mathcal{N}(0, \Sigma_i), \qquad \text{for } i = 1, \ldots, n, \qquad (1)$$

where $\{X_i\}_{i=1}^n \subseteq \mathbb{R}^d$ are the observed data with distinct error distributions, $\{\theta_i^*\}_{i=1}^n$ are the unobserved latent signals drawn from a common unknown prior measure $G^*$, and $\{Z_i\}_{i=1}^n$ are independent Gaussian noise with known positive-definite covariance matrices $\{\Sigma_i\}_{i=1}^n$.

Model (1) falls within the empirical Bayes (EB) framework [Robbins, 1951, 1983], where the goal is to estimate the prior distribution $G^*$ and recover the latent vectors $\{\theta_i^*\}_{i=1}^n$. One way to achieve the goal is through the *G-modeling* approach [Efron, 2014], which is directly modeling the prior on the parameter scale and then plugged into the posterior calculation. A principled and widely used approach to estimate $G^*$, without imposing any parametric assumption on it, is the *nonparametric*

39th Conference on Neural Information Processing Systems (NeurIPS 2025).

*maximum likelihood estimator* (NPMLE) [Robbins, 1950, Kiefer and Wolfowitz, 1956, Jiang and Zhang, 2009, Gu and Koenker, 2016, Polyanskiy and Wu, 2020]. The NPMLE seeks a distribution $\widehat{G} \in \mathcal{P}(\mathbb{R}^d)$ that maximizes the marginal likelihood of the observed data $\{X_i\}_{i=1}^n$:

$$\widehat{G}_{\mathrm{MLE}} \in \underset{G \in \mathcal{P}(\mathbb{R}^d)}{\arg\max} \; \frac{1}{n} \sum_{i=1}^n \log f_{G,\Sigma_i}(X_i), \tag{2}$$

where $\mathcal{P}(\mathbb{R}^d)$ is the family of all probability measures on $\mathbb{R}^d$ and $f_{G,\Sigma_i}(x) := \int \varphi_{\Sigma_i}(x-\theta) dG(\theta)$ is the marginal density of $X_i$ under $G$ with $\varphi_{\Sigma_i}(y) := (\det(2\pi\Sigma_i))^{-1/2} \exp\left(-\frac{1}{2} y^\top \Sigma_i^{-1} y\right)$. Once $\widehat{G}$ is obtained, each latent signal $\theta_i$ can be estimated using the *empirical Bayes* posterior mean:

$$\widehat{\theta}_i := \mathbb{E}_{\widehat{G}}[\,\theta \mid X_i, \Sigma_i\,] \stackrel{(*)}{=} X_i + \Sigma_i \nabla_x \log f_{\widehat{G},\Sigma_i}(X_i), \tag{3}$$

where $(*)$ is due to the Tweedie's formula [Efron, 2011], linking posterior means to $\nabla_x \log f_{\widehat{G},\Sigma_i}(X_i)$, the score function of the marginal density. This nonparametric approach flexibly adapts to latent structures without strong modeling assumptions on $G^*$. Theoretical guarantees for the NPMLE in this model are well established in the homoscedastic setting $\Sigma_i = \Sigma$ or the univariate setting $d = 1$: it achieves a *near-parametric* sample complexity rate $O(n^{-1}(\log n)^{O(d)})$, i.e., parametric up to logarithmic multiplicative factors, for both density estimation and Bayes regret risk [Jiang and Zhang, 2009, Saha and Guntuboyina, 2020]. Recently, the results have further been extended to multivariate and heteroscedastic setting in Soloff et al. [2025].

Despite these guarantees, the NPMLE and its associated Tweedie-based denoising procedure suffer from inherent *statistical and computational limitations* that render their practical effectiveness in high-dimensional settings: (a) Statistically, Tweedie's posterior mean estimator often has an *over-shrinkage bias*. As an example, consider a joint Gaussian model for $(X, \theta^*)$: $\begin{bmatrix} X \\ \theta^* \end{bmatrix} \sim \mathcal{N}\left(\begin{bmatrix} 0 \\ 0 \end{bmatrix}, \begin{bmatrix} 2I_d & I_d \\ I_d & I_d \end{bmatrix}\right)$. It follows that marginally $\theta^* \sim \mathcal{N}(0, I_d)$, and $X|\theta^* \sim \mathcal{N}(\theta^*, I_d)$. However, the posterior mean of $\theta^*$ is $\mathbb{E}[\theta^* | X] = X/2 \sim \mathcal{N}(0, I_d/2)$, thus underestimating the variance of the latent signal. (b) Computationally, although problem (2) is a convex optimization in $G$, it is infinite-dimensional and difficult to solve in practice. Since it is known that there exists a solution $\widehat{G}_{\mathrm{MLE}}$ with support on at most $n$ atoms, one common way to address the issue is to restrict $G$ to be a discrete measure of the form $\sum_{i=1}^m w_i \delta_{\mu_i}$, with $\sum_{i=1}^m w_i = 1$ for some $m \le n$. Optimizing over $\{(w_i, \mu_i)\}_{i=1}^m \subseteq \mathbb{R}_+ \times \mathbb{R}^d$ leads to a finite dimensional but nonconvex optimization, often tackled via the expectation-maximization (EM) algorithm [Laird, 1978]. However, EM suffers from slow convergence and may converge to local maxima due to the nonconvexity. To mitigate these issues, prior work has suggested fixing the atoms $\{\mu_i\}_{i=1}^m$ using either grid points over a compact region [Koenker and Mizera, 2014, Feng and Dicker, 2018] or the observed data itself [Lashkari and Golland, 2007]. Yet the grid-based approach scales poorly with dimension, and the data-driven strategy is suboptimal under heteroscedastic noise [Soloff et al., 2025]. Figure 1 further illustrates the over-shrinkage and computational challenges that arise from these limitations.

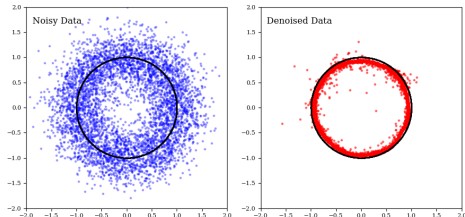 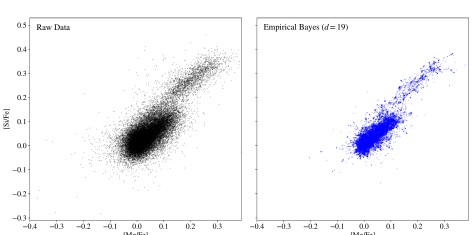

Figure 1: **Left**: 5,000 two-dimensional observations $X_i$ (in blue) sampled from $X_i|\theta_i^* \sim \mathcal{N}(\theta_i^*, (0.3)^2 I_2)$ with $\theta^*$ uniformly distributed over a unit circle (in black). The posterior means (in red), obtained via the NPMLE and (3), are over-shrunk relative to the true unit circle. **Right** (adapted from Soloff et al. [2025, Figure 7]): a real-world 19-dimensional astronomy data (in black). The estimated posterior means (in blue) from NPMLE poorly reveal the underlying chemical structure (see Section 5 for the denoising results from our methods).

## 1.1 Our Methods and Contributions

To address the statistical and computational limitations of the classical NPMLE and Tweedie-based posterior denoising, we propose a novel perspective that reinterprets the data generation mechanism in model (1) through the lens of stochastic diffusion processes. Our method is built on two core ideas: replacing the NPMLE with score matching to estimate the prior within the G-modeling approach, and applying multi-step posterior denoising to recover latent signals. Below, we present an overview of our approach in the simplified homoscedastic setting; the full treatment of the heteroscedastic case is developed in detail in Section 2.

**Modeling the data measurement process.** Consider the homoscedastic version of model (1), where $\Sigma_i = \Sigma$ for all $i = 1, \cdots, n$. In this special case, the observed data $\{X_i\}_{i=1}^n$ are i.i.d. samples from the convolution of the latent distribution $G^*$ with the Gaussian kernel $\mathcal{N}(0, \Sigma)$:

$$X_i \sim G^* * \mathcal{N}(0, \Sigma), \qquad \text{for } i = 1, \ldots, n. \tag{4}$$

Rather than treating each observed data $X_i$ as the result of a static additive noise model, we propose to interpret $\{X_i\}_{i=1}^n$ as snapshots from a forward stochastic diffusion process. Specifically, we model the measurement process as a zero-drift stochastic differential equation (SDE) that continuously perturbs samples from the latent prior $G^*$:

$$dX^{(t)} = g(t)\, \Sigma^{1/2} dB_t, \quad \text{with } X^{(0)} \overset{d}{=} \{\theta_i^*\}_{i=1}^n \sim G^*, \tag{5}$$

where $B_t \in \mathbb{R}^d$ denotes a standard Brownian motion and $g : \mathbb{R}_+ \to \mathbb{R}_+$ is the diffusion coefficient controlling the noise scaling. The function $g(t)$ can be adapted to reflect assumptions on the measurement process; for simplicity, one natural choice is $g(t) = \sqrt{\frac{2}{e^2-1}} e^t \Rightarrow \alpha(t) := \left( \int_0^t g(s)^2 ds \right) = \frac{e^{2t}-1}{e^2-1}$. Under this model, the marginal distribution of $X^{(t)}$ evolves according to:

$$X^{(t)} \sim G^* * \mathcal{N}\left(0, \Sigma^{(t)}\right), \qquad \text{where } \Sigma^{(t)} := \alpha(t)\Sigma = \frac{e^{2t}-1}{e^2-1}\Sigma.$$

In particular, at time $t = 1$, we have $\Sigma^{(1)} = \Sigma$. Therefore, we can indeed view each $X_i$ as sampled from the diffusion process (5) at $t = 1$, i.e., $\{X_i\}_{i=1}^n \sim X^{(1)}$.

**Denoising the observations.** The forward-time process in (5) is associated with a reverse-time diffusion process, which can be approximated by a deterministic flow that shares the same marginal distributions over time [Anderson, 1982, Maoutsa et al., 2020, Song et al., 2021a]. This flow is governed by the following ordinary differential equation (ODE):

$$d\overline{X}^{(t)} = -\frac{1}{2} g(t)^2\, \Sigma\, \nabla_x \log f_{G^*, \Sigma^{(t)}}(\overline{X}^{(t)})\, dt. \tag{6}$$

Given an estimator $\widehat{G}$ of the prior $G^*$, this reverse time ODE naturally defines a generalized *multi-step* Tweedie-type empirical Bayes denoising procedure. Specifically, replacing $G^*$ with $\widehat{G}$ and discretizing the ODE backward in time from $t = 1$ to $t = 0$ with step size $\Delta t$ yields the iterate:

$$\widehat{\theta}_i^{(t-\Delta t)} = \widehat{\theta}_i^{(t)} + \frac{\Delta t}{2}\, g(t)^2\, \Sigma\, \nabla_x \log f_{\widehat{G}, \Sigma^{(t)}}(\widehat{\theta}_i^{(t)}) \quad \text{with initial condition } \widehat{\theta}_i^{(1)} = X_i. \tag{7}$$

The final denoised estimate is taken as $\widehat{\theta}_i := \widehat{\theta}_i^{(0)}$ for $i = 1, \ldots, n$. Notably, this method generalizes Tweedie's formula in (3), which corresponds to a single-step approximation with $\Delta t = 1$. Our multi-step denoising procedure mirrors optimization dynamics, particularly Langevin-type flows over probability spaces, where smaller step sizes yield more controlled updates. This approach allows gradual refinement of the denoised signal, helping mitigate the over-shrinkage bias observed in one-step Tweedie denoising methods.

**Learning the latent prior** $G^*$**.** Our stochastic framework also naturally induces a denoising score matching (DSM) objective [Vincent, 2011] for learning the prior $G^*$. We propose to further perturb the noisy observations $X_i$ using the forward SDE (5) to generate conditional samples from the kernel $p_{1t}(\cdot | X_i) = \varphi_{\Sigma^{(t)} - \Sigma}(\cdot - X_i)$, up to some stopping time $T$. We then match the score of the model's marginal density to the conditional score induced by this perturbation:

$$\widehat{G} := \operatorname*{arg\,min}_{G \in \mathcal{P}(\mathbb{R}^d)} \frac{1}{n} \sum_{i=1}^n \int_1^T \mathbb{E}_{p_{1t}(\cdot|X_i)} \left[ g(t)^2 \left\| \nabla \log f_{G, \Sigma^{(t)}}(x) - \nabla \log p_{1t}(x|X_i) \right\|_\Sigma^2 \right] dt, \tag{8}$$

where $\|x\|_\Sigma^2 := x^\top \Sigma\, x$ denotes a weighted $\ell_2$-norm, extending the likelihood-weighting idea of Song et al. [2021b]. Intuitively, under our stochastic view, the observations at $t = 1$ already contain calibrated noise relative to $G^*$, and additional perturbation simulates conditional transitions. Matching model scores to these local conditional scores allows global recovery of the latent prior. Unlike standard diffusion models with the DSM objective that directly parameterize the score function with neural networks [Song et al., 2021a], we model the distribution $G$ explicitly in (8). Such $G$-modeling is crucial in deriving new theoretical guarantees for the score estimation risk. Computationally, although the objective 8 is nonconvex, diffusion-type squared loss objectives have been empirically observed to exhibit smoother optimization landscapes and avoid poor local minima [Xu et al., 2024]. This empirical tractability is further supported by theory on the benign optimization behavior of overparameterized neural networks trained over mean-squared losses that can typically satisfy the Polyak–Łojasiewicz condition or behave linearly in the neural tangent kernel regime [Liu et al., 2022].

In summary, we propose a novel integration of score-based diffusion models with the classical EB framework via G-modeling. The rest of the paper is organized as follows: In Section 2, we present a stochastic process framework to model heteroscedastic Gaussian mixtures, where each observation is treated as a marginal realization from a distinct diffusion process with its own score function. Section 3 establishes the theoretical advantages of our approach, showing that G-modeling combined with diffusion yields near-parametric rates for score estimation. We further discuss how to numerically solve the $G$-modeling based score matching objective in Section 4. In Section 5, we present experimental results on both synthetic and real astronomical data, demonstrating substantial improvements in denoising and prior recovery over NPMLE and one-step Tweedie-based methods, supporting the improved computational scalability of our framework in high-dimensional settings.

## 1.2 Related Work

**Score-based models for mixtures.** Recent theory has provided provable guarantees for diffusion and score-based models in learning Gaussian mixtures, yet these results neither handle the heteroscedastic measurement setting we study nor aim at recovering the latent prior itself. Works such as Gatmiry et al. [2024], Chen et al. [2024], Shah et al. [2023] propose diffusion-based algorithms for mixture models under either finite discrete structures or continuous latent mixtures with homoscedastic noise. These approaches typically assume access to an infinite supply of i.i.d. samples and yield slow convergence guarantees of order $O((\log n)^{-1/p})$ in score or total variation distance, for some exponent $p > 1$. Importantly, these works directly parametrize the score function, using either neural networks or piecewise polynomial surrogates, without attempting nonparametric recovery of the latent mixing measure $G^*$ or incorporate explicit G-modeling into the objective. Because of this direct score parametrization, these methods cannot be readily applied to our core task of latent recovery and are incompatible with our de-biased multi-step denoising procedure. A notable exception is Ghosh et al. [2025], which presents the first analysis of combining score matching with G-modeling, achieving a near-parametric rate $O(\log^6 n/n)$ for score estimation risk. However, their results are limited to the univariate, homoscedastic setting ($d = 1$). In contrast, our work tackles the more challenging regime of multivariate, heteroscedastic Gaussian location mixtures with arbitrary dimension $d \geq 1$. By coupling G-modeling with a stochastic process perspective, we develop an explicit estimator of the latent prior $G^*$ and establish near-parametric rates for score estimation (up to small smoothing), along with finite-sample guarantees for recovering the latent signals.

**Image denoising in computer vision.** While our problem and the proposed method may appear superficially related to score-based diffusion approaches for image denoising, the settings and statistical challenges are fundamentally different. Most notably, our task lacks access to clean ground truth data, precluding the use of pre-trained score models, and all inference must be performed from noisy observations alone. In contrast, supervised denoising approaches such as BM3D [Dabov et al., 2006] requires access to a large number of paired noisy-clean images, while self-supervised methods like Noise2Score [Kim and Ye, 2021] rely on one-step Tweedie approximations that ignore the unknown prior $G^*$ and leverage only partial model information $\Sigma_i$. These methods are either unrealistic or statistically inefficient in scientific domains where latent structure is present but clean data are unavailable, such as the astronomy measurement data. Even in moderate dimensions (e.g., $d = 20$), empirical Bayes is a highly nontrivial statistical task due to the data sparsity: each observation $X_i$ corresponds to a unique latent parameter $\theta_i^*$ and an individual covariance $\Sigma_i$, yielding just one single noisy sample to infer each $\theta_i^*$.

## 2 SDGM: Score-based Diffusion with G-Modeling

In this section, we present our general treatment of the score-based diffusion approach with G-modeling under the heteroscedastic setting. The key idea is still to reinterpret each noisy observation as a time-1 snapshot from a latent stochastic diffusion process. While our homoscedastic formulation serves as a motivating special case, the heteroscedastic setting introduces a crucial distinction: each observation $X_i$ corresponds to a different SDE, despite all sharing a common latent initialization. Specifically, we consider the following family of zero-drift linear SDEs:

$$dX_i^{(t)} = g_i(t)\Sigma_i^{1/2}dB_t, \quad \text{with } X_i^{(0)} \sim G^*, \qquad \text{for } i = 1,\ldots,n, \tag{9}$$

where $\{g_i(t)\}_{i=1}^n \subset \mathbb{R}$ are the time-dependent diffusion coefficients. Without loss of generality, we assume $\alpha_i(t) := \int_0^t g_i(s)^2 ds$ satisfies $\alpha_i(1) = 1$ for all $i$. Then the time-dependent covariances are given by $\Sigma_i^{(t)} := \alpha_i(t)\,\Sigma_i$, and the marginal law of $X_i^{(t)}$ becomes

$$X_i^{(t)} \sim G^* * \mathcal{N}(0, \Sigma_i^{(t)}), \qquad \text{for } t > 0, \ i = 1,\ldots,n. \tag{10}$$

Thus, each observation $X_i$ in the original mixture model (1) can be viewed as a marginal sample from this diffusion process at $t = 1$, i.e., $X_i \stackrel{d}{=} X_i^{(1)} \sim G^* * \mathcal{N}(0, \Sigma_i)$ for all $i = 1,\ldots,n$.

As in the homoscedastic case, each forward SDE in (9) is coupled with the reverse-time ODE:

$$d\overline{X}_i^{(t)} = -\frac{1}{2}g_i(t)^2 \cdot \Sigma_i \cdot \nabla_x \log f_{G^*,\Sigma_i^{(t)}}(\overline{X}_i)\,dt, \qquad \text{for } i = 1,\ldots,n.$$

Given any plug-in estimator $\widehat{G}$ for $G^*$, substituting $\Sigma^{(t)} = \Sigma_i^{(t)}$ in (7) defines a multi-step score-based denoising procedure that remains consistent with the EB framework. Furthermore, as $\Delta t \to 0$, the dynamics in (7) induces a deterministic transport map from the noisy empirical measure toward a latent measure $\widehat{G} \approx G^*$. This perspective aligns with the design principles behind recent optimal transport approaches for empirical Bayes denoising [Soloff et al., 2025, Zhang et al., 2024], mitigating the over-shrinkage limitation of one-step Tweedie-type procedures.

Importantly, in our setting, the SDEs in (9) serve as a *belief model* for the true data-generating process, and the noisy observations in (1) are treated as marginal realizations from latent diffusion paths. This fundamentally differs from the generative score-based diffusion models like Song et al. [2021a], where SDEs are artificially constructed for the purpose of noise injection and score network training. In an idealized setting where the true latent measure $G^*$ is known or accessible via unlimited samples, one could apply the framework of Song et al. [2021b,a] to learn a time-dependent score network $s_\theta(\cdot, t)$, mapping $G^*$ to pure noise through an invertible path. However, our setting departs significantly from this setup. First, we observe only a finite collection of noisy samples at forward time $t = 1$, each generated by a unique, heteroscedastic SDE rather than homogeneous, artificially perturbed noise. Second, our goal is to explicitly estimate the latent distribution $G^*$, as opposed to modeling it implicitly via score functions. Finally, because no data is available at intermediate times $t \in (0, 1)$, the standard time-indexed score network $s_\theta(\cdot, t)$ cannot be trained in our setting.

To address these challenges, we propose a new objective that aggregates information across all diffusion paths to estimate $G^*$. Our method, which we term *score-based diffusion with G-modeling* (SDGM), obtains the estimator $\widehat{G}$ by:

$$\widehat{G} \in \underset{G \in \mathcal{P}(\mathbb{R}^d)}{\arg\min} \mathcal{J}_n(G) := \frac{1}{n}\sum_{i=1}^n \int_{t_0}^T g_i(t)^2 \mathbb{E}_{x\,|\,i,t}\left[\left\|\nabla \log q_G^{(t)}(x) - \nabla \log q_n^{(t)}(x)\right\|_{\Sigma_i}^2\right] dt, \tag{11}$$

where, $t_0 \geq 1$ is the minimum diffusion time, and to simulate the homoscedastic behavior, we consider the *rescaled* Gaussian transition kernel: $x\,|\,i, t \sim K_t(\cdot\,|\,i) := \mathcal{N}([\alpha_i(t)\Sigma_i]^{-1/2}X_i, (1 - \alpha_i(t)^{-1})I_d)$ defined for $t \geq t_0 \geq 1$, and define the mixture densities $q_n^{(t)}$ and $q_G^{(t)}$ to be:

$$\text{(empirical) } q_n^{(t)} := \frac{1}{n}\sum_{i=1}^n K_t(\cdot\,|\,i), \qquad \text{(G-induced) } q_G^{(t)} := \frac{1}{n}\sum_{i=1}^n f_{(\Sigma_i^{(t)})^{-1/2}\#G, I_d},$$

where $T\#G$ denotes the pushforward measure of $G$ under map $T$ such that for every set $A \subseteq \mathbb{R}^d$, $(T\#G)(A) = G(T^{-1}A)$. The defined mixtures satisfy that $\mathbb{E}q_n^{(t)} = q_{G^*}^{(t)}$. Intuitively, $q_G^{(t)}$ aggregates

the global behavior of the latent measure $G$ across all SDEs, while $q_n^{(t)}$ serves as its empirical counterpart based on noisy observations. By matching the score functions of these two mixtures, we guide $\widehat{G}$ toward an accurate approximation of $G^*$. In the homoscedastic special case with $\Sigma_i = \Sigma$ and $g_i = g$, the mixtures simplify to a rescaled version of the objective (11) that is equivalent to the score matching formulation in (8) up to constants.

# 3  Theoretical Guarantees

In this section, we provide guarantees on the statistical performance for both homoscedastic and heteroscedastic noise models. We primarily study three statistical properties to quantify how well the SDGM estimator $\widehat{G}$ of (11) performs as a plug-in estimator for the true prior $G^*$.

The successful convergence of diffusion-based generative models is underlaid by a low score estimation error [Yakovlev and Puchkin, 2025, Bortoli, 2022]. Define the following *time-cumulative average Fisher divergence* for all path measures $\mu$ and $\nu$:

$$\overline{\mathfrak{F}}_{[t_0,T]}(\mu||\nu) := \frac{1}{n}\sum_{i=1}^{n}\int_{t_0}^{T}\mathbb{E}_{x\sim\mu_i^{(t)}}\|\nabla\log\mu_i^{(t)} - \nabla\log\nu_i^{(t)}\|_2^2 dt.$$

Then, the specialized metric $\overline{\mathfrak{F}}_{[t_0,T]}(f_{G^*,\Sigma^{(t)}}||f_{\widehat{G},\Sigma^{(t)}})$ measures the time-cumulative risk on how $\nabla\log f_{G,\Sigma^{(t)}}$ approximates $\nabla\log f_{G^*,\Sigma^{(t)}}$ along the forward-time SDE path beyond the observable time. It is consistent with the generalization risk concerned in Yakovlev and Puchkin [2025].

For density estimation, $\widehat{G}$ also provides a natural estimate for the mixture marginal law $q_{G^*}^{(t)}$ aggregated from the paths of (9). We measure the quality of density estimation by the standard squared Hellinger accuracy, $\mathfrak{H}^2(p,q) := \frac{1}{2}\int(\sqrt{p} - \sqrt{q})^2$ between densities $p$ and $q$. In particular, we are interested in the following squared Hellinger distances:

$$(\text{homoscedastic})\ \mathfrak{H}^2(f_{G^*,\Sigma^{(t)}}, f_{\widehat{G},\Sigma^{(t)}}), \qquad (\text{heteroscedastic})\ \mathfrak{H}^2(q_{G^*}^{(t)}, q_{\widehat{G}}^{(t)}),$$

which captures the global fit of the plug-in estimator $\widehat{G}$ for estimating the joint law of $\{X_i^{(t)}\}_{t\geq 1}$.

The third metric of interest we establish is the squared Wasserstein distance (see e.g., Villani [2008]), which is defined for all $\mu, \nu \in \mathcal{P}(\mathbb{R}^d)$ that $W_2^2(\mu,\nu) := \inf_{\gamma\in\Pi(\mu,\nu)}\mathbb{E}_{(x,y)\sim\gamma}[\|x-y\|_2^2]$ with $\Pi(\mu,\nu)$ denotes the set of all joint measures over $\mu,\nu$. The distance $W_2^2(\widehat{G}, G^*)$ naturally characterizes the direct consistency of $\widehat{G}$ for the true $G^*$.

We make the following assumptions on the latent structure and the data generation process (9). In the assumptions, the support conditions are widely considered in literature [Jiang and Zhang, 2009, Saha and Guntuboyina, 2020, Ghosh et al., 2025]. They encompass a wide range of realistic scenarios, including discrete mixtures, compactly supported manifold-supported latent structures, and representative experiment settings in Section 5. Furthermore, well-conditioned assumptions on noise covariances are ubiquitous in prior works [Soloff et al., 2025] and are essential to ensure smoothly-defined score everywhere and to enable information-theoretically meaningful latent recovery.

**Assumption 1.** *The true latent measure in model* (1) *is compactly supported, i.e.,* $G^* \in \mathcal{P}([-M,M]^d)$ *for some finite radius* $M > 0$. *There exists* $0 < \underline{\sigma} \leq \overline{\sigma} < \infty$ *such that the covariances satisfy* $\underline{\sigma}I_d \preceq \Sigma_i \preceq \overline{\sigma}I_d$ *for all* $i = 1, \ldots, n$. *The diffusion coefficient satisfies* $\alpha_i(1) = 1$ *(such that* $\Sigma_i^{(1)} = \Sigma_i$*) and other mild regularity conditions specified in the supplemental.*

Our first theorem characterizes the performance of our SDGM framework for the canonical homoscedastic model (4), which is specialized from our guarantee for the heteroscedastic model.

**Theorem 1** (Homoscedastic noise). *Under Assumption 1 and further assume that* $\Sigma_i = \Sigma$, $g_i = g$ *for all* $i = 1, \ldots, n$. *Let* $\widehat{G}$ *minimizes objective* (8) *constrained over the family* $\mathcal{P}([-M,M]^d)$ *and with parameters* $t_0$ *and* $T$ *satisfying* $1 + \frac{1}{\log n} \leq \alpha(t_0) \leq \alpha(T) \leq 2$. *Then provided* $n$ *sufficiently large, the following bounds hold with probability at least* $1 - n^{-2}$:

*Score and density estimation:*

$$\mathbb{E}\overline{\mathfrak{F}}_{[t_0,T]}(f_{G^*,\Sigma^{(t)}}\|f_{\widehat{G},\Sigma^{(t)}}) \lesssim_{M,d,(\underline{\sigma},\overline{\sigma}),g} \frac{1}{n}(\log n)^{2d+3}, \tag{12}$$

$$\mathbb{E}\mathfrak{H}^2(f_{G^*,\Sigma^{(t_0)}}, f_{\widehat{G},\Sigma^{(t_0)}}) \lesssim_{M,d,(\underline{\sigma},\overline{\sigma}),g} \frac{1}{n}(\log n)^{2d+3}. \tag{13}$$

*Deconvolution:*

$$\mathbb{E}W_2^2(G^*,\widehat{G}) \lesssim \frac{1}{\log n}. \tag{14}$$

Here, we write $f(n) \lesssim g(n)$ if there exists a universal constant $C$ independent of $n$ such that $f(n) \le Cg(n)$, $\forall n$. We write $\lesssim_{\theta_1,\theta_2,\ldots}$ to emphasize that the constant $C = C_\theta$ also depends on the fixed specification parameters $\theta_i$'s. In the following, we discuss implications of this result. Under a latent manifold setting with homoscedastic isotropic noise and neural parameterization of the score function, recent work [Yakovlev and Puchkin, 2025] proves the rate $\mathbb{E}\overline{\mathfrak{F}}_{[t_0,T]}(f_{G^*,\Sigma^{(t)}}\|f_{\widehat{G},\Sigma^{(t)}}) = O(n^{-2\beta/(4\beta+d_0)}\mathrm{polylog}(n))$, where $d_0$ is the intrinsic dimension of the manifold and $\beta$ denotes a smoothness parameter. To our knowledge, this is the sharpest rate known for diffusion-based score estimation under such assumptions. By leveraging explicit $G$-modeling to impose the structure on the estimated scores, our result achieves an improved $O(n^{-1})$ leading dependence on sample size $n$ (up to logarithmic factors), under more general assumptions. Moreover, our Hellinger accuracy bound (13), modulo a small distributional shift from $t = 1$ to $t_0$, matches the rate achieved in NPMLE literature [Saha and Guntuboyina, 2020]. If the target is to estimate $f_{G^*,\Sigma}$ at the observational noise level, an unavoidable smoothing cost due to diffusion arises. In this regime, the task incurs the inevitable minimax rate $O(n^{-2/(d+4)})$ that cannot compete with NPMLE [Wibisono et al., 2024]. Finally, in the Gaussian deconvolution setting with fixed noise design, the minimax rate deteriorates to the logarithmic regime $O(1/\log n)$ due to inherent ill-posedness induced by Gaussian smoothing and it is the best rate possibly achieved by both NPMLE and our framework [Soloff et al., 2025, Dedecker and Michel, 2013]. We further discuss limitations of the current theory in Section 6.

Our next theorem generalizes the results to the heteroscedastic settings. The core contents of Theorem 2 parallels that of Theorem 1, but additional care is needed to account for observation-specific noise covariances and SDEs. To make the readers easy to follow, we present the main result informally below and defer detailed technical derivations and proofs of Theorem 1 and 2 to the supplemental material.

**Theorem 2** (Heteroscedastic noise; Informal). *Under Assumption 1, provided solution $\widehat{G}$ to (11) constrained over the family $\mathcal{P}([-M,M]^d)$ with $(t_0,T)$ satisfying well-behaved smoothing condition, then with high probability,*

$$\mathbb{E}\overline{\mathfrak{F}}_{[t_0,T]}(q_{G^*}^{(t_0)}\|q_{\widehat{G}}^{(t_0)}) \lesssim \frac{(\log n)^{2d+3}}{n}, \qquad \mathbb{E}\mathfrak{H}^2(q_{G^*}^{(t_0)}, q_{\widehat{G}}^{(t_0)}) \lesssim \frac{(\log n)^{2d+3}}{n},$$

$$\mathbb{E}W_2^2\left(\frac{1}{n}\sum_{i=1}^{n}(\Sigma_i)^{-1/2}\#G^*, \frac{1}{n}\sum_{i=1}^{n}(\Sigma_i)^{-1/2}\#\widehat{G}\right) \lesssim \frac{1}{\log n}.$$

# 4 Solution Methods to Solve (11)

The optimization problem in (11) is an *infinite dimensional nonconvex* problem that is challenging to solve numerically. We discuss practical strategies for approximately computing the SDGM solution.

**Strategy one: discretize the support of $G$.** A classical approach to deal with the infinite dimensional variable $G$, as commonly used in NPMLE, is to restrict the class of priors to discrete measures with a finite number of supports: $G = \sum_{j=1}^{m} w_j \delta_{\mu_j}$, with $\sum_{j=1}^{m} w_j = 1$, $w_j \ge 0$. This reduces the optimization to a finite-dimensional, albeit nonconvex, problem over $\{(w_j, \mu_j)\}_{j=1}^{m}$.

**Strategy two: continuous modeling with normalizing flows.** An alternative approach is to model $G$ continuously using normalizing flows [Rezende and Mohamed, 2015, Dinh et al., 2017, Durkan et al., 2019]. A normalizing flow represents a flexible distribution as the pushforward of a simple base distribution (e.g., standard Gaussian) through an invertible neural network. We parameterize $G$

using a Neural Spline Flow [Durkan et al., 2019] implemented via the `nflows` library [Durkan et al., 2020], allowing efficient sampling and density evaluation. In our framework, we draw Monte Carlo samples from the flow to estimate $f_G$ and its score during training. Sampling from a continuous prior is theoretically justified by empirical Bayes approximation results: given a continuous $G$, there exists a discrete approximation $G'$ that closely matches $f_G$ and its gradient [Saha and Guntuboyina, 2020, Lemma D.3]. Furthermore, continuous modeling enables richer recovery of the underlying structure of $G$, which is crucial for scientific interpretation and denoising applications [Shen and Wu, 2022].

**Batched and conditional estimation of $q_G$.** The full mixture $q_G(x)$ plays a central role in our theoretical guarantees, but evaluating its score $\nabla_x \log q_G$ is computationally intensive for large $n$. A natural relaxation is to approximate $q_G$ at each iteration using only a mini-batch of observations. In particular, we focus on the case where the mini-batch size $B = 1$: at each training step, we align the rescaled conditional score $\nabla_x \log f_{G,\Sigma_i^{(t)}}(x)$ with the empirical conditional score $\nabla_x \log \mathcal{N}(x; X_i, (\alpha_i(t) - 1)\Sigma_i)$. This corresponds to a *conditional score matching* update, mirroring the classical intuition of denoising score matching [Vincent, 2011], while offering computational simplicity and stability in stochastic optimization. In fact, subsampled estimation for mixture inference has also been used in early works [Celeux et al., 2001, Ihler et al., 2003].

## 5 Numerical Experiments

In this section, we evaluate the empirical performance of our proposed method on synthetic heteroscedastic datasets and real-world astronomy data. We also investigate whether the empirical scaling behavior aligns with our theoretical guarantees, particularly as we vary the problem dimension. We refer the readers to the supplemental for more implementation details and results.

**Baselines and metrics.** We evaluate two variants of our proposed method: SDGM-C, which uses a continuous prior modeled via normalizing flows, and SDGM-D, which uses a discrete prior. We compare our approaches against the state-of-the-art NPMLE solvers for the problem implemented in literature. Specifically, we compare with (1) NPMLE-N, the discretized NPMLE with supports taken fixed at noisy data (thus the objective (2) is optimized over only weights and becomes convex), as advocated in Lashkari and Golland [2007]; we adopt the Newton-based augmented Lagrangian method implemented from [Zhang et al., 2024]. (2) PEM, the partial EM method which is a generalized EM method proposed in Zhang et al. [2022] that used tighter sequential relaxations.

In addition, we consider a hybrid heuristic called SDGM-P to further improve the likelihood value defined in (2). This method first trains the SDGM-C model to learn a continuous prior, from which we sample support points. These support points are then used for a *post-processing* step via the convex NPMLE-N solver, effectively refining the prior estimate to maximize the likelihood.

We evaluate the performance of different methods over following metrics: (1) Fisher divergence (FD) defined by $\mathfrak{F}_n(f_G, f_*) := \frac{1}{n} \sum_{i=1}^n \|\nabla \log f_{G^*,\Sigma_i}(X_i) - \nabla \log f_{\widehat{G},\Sigma_i}(X_i)\|_2^2$. (2) Negative log-likelihood (NLL) defined by $-\frac{1}{n} \sum_{i=1}^n \log f_{\widehat{G},\Sigma_i}(X_i)$, which serves as a proximal for the KL divergence between $f_{\widehat{G},\Sigma_i}$ and $f_{G^*,\Sigma_i}$ (3) Squared Wasserstein-2 distance ($W_2^2$) between estimated $\widehat{G}$ and $G^*$, approximated by the entropy-regularized optimal transport distance [Cuturi, 2013, Genevay et al., 2018]. (4) Mean squared error (MSE) between the denoised and clean data $\frac{1}{n} \sum_{i=1}^n \|\widehat{\theta}_i - \theta_i^*\|_2^2$.

**Synthetic datasets.** We evaluate our method on two synthetic datasets with low-dimensional manifold structure, both of which are widely used in the NPMLE literature [Saha and Guntuboyina, 2020, Zhang et al., 2024]. Specifically, for $d \geq 2$, we generate clean latent vectors $\theta_i^*$ that exhibit nontrivial structure only in the first two dimensions, with the remaining dimensions padded with zeros:

Example 1 (circle): The first two coordinates of each $\theta_i^*$ are drawn uniformly from a circle of radius 6 centered at the origin.

Example 2 (uppercase letter "A"): The first two coordinates of each $\theta_i^*$ are drawn uniformly from one of five line segments that form the shape of the capital letter "A." These segments connect the points $(-4, 6)$, $(-2, 0)$, $(0, 6)$, $(2, 0)$, and $(4, 6)$, with each segment selected with equal probability $1/5$.

We then corrupt each $\theta_i^*$ with independently generated heteroscedastic Gaussian noise $Z_i \sim \mathcal{N}(0, \Sigma_i)$, where each $\Sigma_i$ is a diagonal matrix. The diagonal entries of $\Sigma_i$ are sampled independently and uniformly from the interval $[(0.8)^2, (1.2)^2]$.

Table 1: Models are trained with dimension $d = 16$ and sample size $n = 2 \times 10^4$. The ↓ indicates that lower values are better. We report generalization risks evaluated on a separate test dataset of size $10^4$. Denoising of the baseline methods NPMLE-N and PEM is performed via the one-step Tweedie's formula, while our methods use the ODE solver. Results for stochastic methods are averaged over 5 independent runs.

| $G^*$ | circle | | | | uppercase letter "A" | | | |
|---|---|---|---|---|---|---|---|---|
| metric | FD ↓ | NLL ↓ | $W_2^2$ ↓ | MSE ↓ | FD ↓ | NLL ↓ | $W_2^2$ ↓ | MSE ↓ |
| **NPMLE-N** | 46.82 | 25.88 | 14.43 | 3.57 | 34.88 | 25.10 | 12.41 | 3.04 |
| **PEM** | 4.71 | 24.89 | 4.52 | 1.26 | 3.78 | 24.40 | 4.09 | 1.35 |
| **SDGM-C** | $1.27_{\pm 0.04}$ | $24.82_{\pm 0.01}$ | $2.07_{\pm 0.77}$ | $\mathbf{1.13}_{\pm 0.01}$ | $1.23_{\pm 0.11}$ | $24.33_{\pm 0.01}$ | $1.68_{\pm 0.31}$ | $\mathbf{1.32}_{\pm 0.03}$ |
| **SDGM-D** | $3.72_{\pm 0.05}$ | $24.88_{\pm 0.00}$ | $3.95_{\pm 0.03}$ | $1.64_{\pm 0.03}$ | $5.26_{\pm 0.05}$ | $24.41_{\pm 0.00}$ | $3.94_{\pm 0.00}$ | $1.73_{\pm 0.02}$ |
| **SDGM-P** | $\mathbf{0.68}_{\pm 0.06}$ | $\mathbf{24.78}_{\pm 0.00}$ | $\mathbf{1.07}_{\pm 0.06}$ | $1.29_{\pm 0.02}$ | $\mathbf{0.49}_{\pm 0.02}$ | $\mathbf{24.30}_{\pm 0.00}$ | $\mathbf{1.25}_{\pm 0.07}$ | $1.45_{\pm 0.02}$ |

The results are summarized in Table 1 and Figures 2 and 3, where $\widehat{G}$ of our methods are plotted using the marginal density estimated through geodesic-KDE [Vincent and Bengio, 2002]. As shown in Table 1, SDGM consistently outperforms the NPMLE baselines across all metrics, with the additional post-processing step (SDGM-P) further improving the quality of prior estimation in most cases. The denoising effects of different methods are evident in Figures 2 and 3. Specifically, the estimated signals $\widehat{\theta}_i$ from NPMLE and one-step Tweedie denoising appear overly blurred and shrunk, failing to recover the underlying manifold structure. In contrast, SDGM, equipped with multi-step ODE-based denoising, is able to clearly capture and preserve the latent geometry of the data.

**Real-world astronomy data.** We further demonstrate our SDGM-C on a challenging real-world scientific problem: estimating the latent distribution of stellar chemical abundances and denoising observations from the Apache Point Observatory Galactic Evolution Experiment (APOGEE) Data Release 14 [Abolfathi et al., 2018, Majewski et al., 2017]. With pre-processing in Ratcliffe et al. [2020], this dataset, comprising $n = 27,238$ stars with $d = 19$ chemical abundance measurements (e.g., [Fe/H], [Mg/Fe], [Si/Fe]), features heteroscedastic, element-correlated Gaussian noise for each star. The true underlying abundances $\theta_i^*$ are expected to reside on a lower-dimensional manifold governed by astrophysical nucleosynthetic processes [Ting et al., 2012, Weinberg et al., 2019]. Accurately revealing the underlying physical structures through the denoised abundances is crucial for reliable studies of Milky Way formation and evolution [Freeman and Bland-Hawthorn, 2002]. As depicted earlier in Figure 1, applying the NPMLE across the full 19-dimensional space is computationally prohibitive and produces unreliable estimates, as evidenced by Soloff et al. [2025], Zhang et al. [2024]. Notably, applying SDGM-C to the APOGEE data (Figure 4) reveals sharpened astrochemically significant structures. Taken together with our theoretical guarantees, these empirical results showcase that SDGM couples superior denoising fidelity with graceful high-dimensional scalability, making it a promising tool for robust inference in complex scientific datasets.

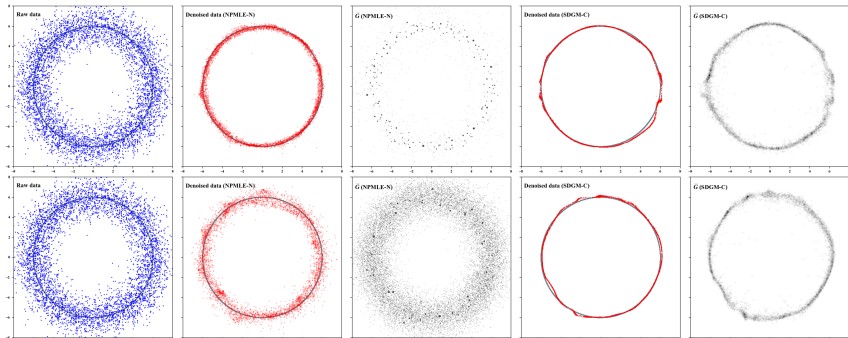

Figure 2: Projected plots of the circle experiment onto the first two dimensions. NPMLE-N (columns 2 & 3) performance significantly degrades when the dimension increases from $d = 8$ (top row) to $d = 16$ (bottom row). Our proposed SDGM-C (columns 4 & 5) stays efficient when $d$ grows.

# 6 Discussions, Limitations, and Future Work

This work develops a stochastic-process reinterpretation of Gaussian location mixtures that bridges classical empirical Bayes inference and modern diffusion-based modeling. Through this lens, we introduce a diffusion-type $G$-modeling objective and a multi-step generalized Tweedie denoising procedure that jointly estimate the latent measure and recover the underlying signals from het-

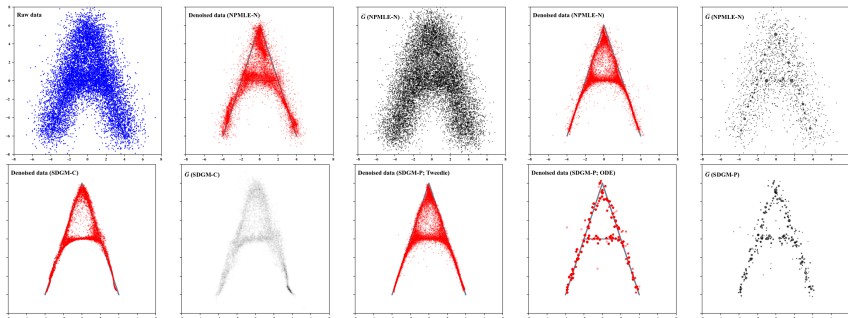

Figure 3: Projected plots of the letter $A$ experiment in $d = 16$ onto the first two dimensions. NPMLE and PEM (top row) struggled by learning only a noisy latent prior, and causing denoised data to collapse within the region between the two legs of the $A$, illustrating over-shrinkage. The bottom row shows our SDGM variants. Notably, even $\widehat{G}$ is estimated by SDGM, the one-step Tweedie denoising (middle of the bottom row) still over-shrinks $\theta_i$. By contrast, the proposed denoising scheme better preserves the $A$ structure, with fewer points confined to the center of the $A$.

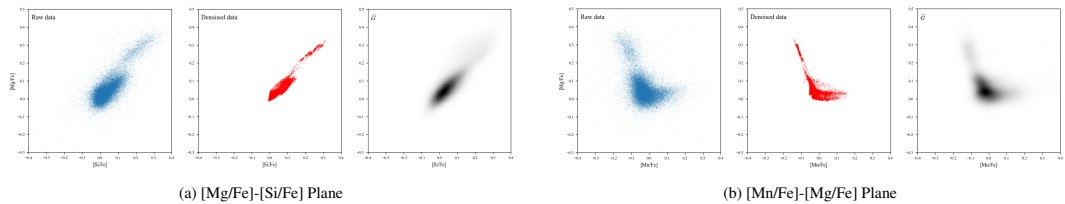

(a) [Mg/Fe]-[Si/Fe] Plane                                   (b) [Mn/Fe]-[Mg/Fe] Plane

Figure 4: **Left:** Noisy APOGEE observations projected onto two 2-D planes. **Middle:** The denoised data reveals a fusiform-like or banana-like structure in the central region and a thin manifold structure in the upper region. **Right:** The marginal density of fitted $\widehat{G}$ suggests strong abundance correlation among the chemicals. These results significantly improve the performance of NPMLE and Tweedie denoising in Figure 1.

eroscedastic mixtures. Unlike standard score-based diffusion approaches that directly parameterize the score function, our method explicitly models the latent prior distribution $G$, which is crucial for consistent recovery of the latent law. In contrast to traditional NPMLE solvers that suffer from slow convergence and over-shrinkage bias, our framework achieves comparable or improved denoising and prior recovery performance, with better empirical scalability in high-dimensional regimes.

**Limitations.** Our theoretical analysis builds on assumptions comparable to those in prior NPMLE studies [Saha and Guntuboyina, 2020, Soloff et al., 2025] and attains a near-parametric rate of $n^{-1}(\log n)^{O(d)}$. These results rely on standard regularity conditions, including compact support of the latent measure and well-conditioned covariance matrices, which are common in NPMLE theory but restrict general applicability in heavy-tailed or ill-conditioned settings. Although the rate matches the minimax lower bound in Hellinger distance for Gaussian mixture estimation [Kim and Guntuboyina, 2022], the dependence on the ambient dimension $d$ remains unfavorable and limits its practicality in very high dimensions. This reflects the intrinsic difficulty of estimating a $d$-dimensional latent measure and recovering each $\theta_i$ from a single noisy sample $X_i$. Since our experiments show that the score-based $G$-modeling approach is more robust and computationally scalable than existing NPMLE solvers, and diffusion models have been observed to capture low-dimensional data manifolds [Pidstrigach, 2022], a promising future direction is to derive rates depending only on the intrinsic dimension $d_0 \ll d$ of the manifold supporting $G^*$, following assumptions similar to Yakovlev and Puchkin [2025]. Algorithmically, our current solution methods rely on heuristic sampling and approximations that do not guarantee convergence to global minima of the infinite-dimensional nonconvex objective. The formulation in Section 2 was primarily chosen to ensure theoretical tractability under the heteroscedastic setting and is not yet optimized for large-scale computation. Future work should aim to bridge this gap between theory and implementation by refining both the modeling formulation and algorithmic design, supported by further analysis of optimization and approximation errors within the empirical Bayes framework.

**Broader implications.** The line of work suggests that integrating diffusion-based generative modeling with nonparametric empirical Bayes principles can yield a unified approach to statistical deconvolution, manifold recovery, and inverse inference under heteroscedastic noise. Future investigations can further extend this framework toward manifold-aware score estimation and other related directions that enhance its theoretical scope and practical applicability.

## Acknowledgment

The authors are partially supported by the National Science Foundation Division of Mathematical Sciences [Grant DMS-2416250] and the National Institutes of Health [Grant 1R01CA287413-01].

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

# Supplemental Material for "Score-Based Diffusion Modeling for Nonparametric Empirical Bayes in Heteroscedastic Gaussian Mixtures"

## A  Code

Our code is available at `https://github.com/chgongyu/sdgm-neurips.git`.

## B  Proofs of Results in Section 3

### B.1  Formal statements of Theorem 2

**Assumption 2.** *For the SDEs defined in* (9)*, assume*

1. *(latent measure)* $G^* \in \mathcal{P}([-M, M]^d)$ *for some radius* $M \in [0, \infty)$.

2. *(covariances)* $\underline{\sigma} I_d \preceq \Sigma_i \preceq \overline{\sigma} I_d$ *for all* $i = 1, \ldots, n$.

3. *(diffusion coefficients) For all* $i = 1, \ldots, n$, $g_i : \mathbb{R}_+ \to \mathbb{R}$ *is a continuous function. Let* $\alpha_i(t) := \int_0^t g_i(s)^2 ds$. *We require the normalization condition* $\alpha_i(1) = 1$, *and additionally the well-behavior condition: fix a* $T > 1$, *then for every* $t \in [1, T]$, *there exists* $(\underline{g}, \overline{g}) > 0$ *such that* $\underline{g} \leq |g_i(t)^2| \leq \overline{g}$.

**Theorem 3.** *Provided conditions in Assumption 2, let* $(t_0, T)$ *satisfy* $t_0 \leq T$ *and*

$$\frac{1}{\log n} \leq \int_1^{t_0} g_i^2(t)dt \leq \int_1^T g_i^2(t)dt \leq 1.$$

*Let* $\widehat{G}_n$ *be an optimal solution to Objective* (11) *constrained over the measure class* $\mathcal{P}([-M, M]^d)$, *then provided* $n$ *sufficiently large, with probability at least* $1 - n^{-2}$,

- *for score estimation:*

$$\mathbb{E}\overline{\mathfrak{F}}_{[t_0, T]}(q_{G^*}^{(t)}||q_{\widehat{G}_n}^{(t)}) := \int_{t_0}^T \mathbb{E}_{x \sim q_{G^*}^{(t)}} \|\nabla \log q_{G^*}^{(t)}(x) - \nabla \log q_{\widehat{G}_n}^{(t)}(x)\|_2^2 dt$$

$$\leq C_{d,M,(\overline{\sigma},\underline{\sigma}),(\underline{g},\overline{g})} \frac{1}{n}(\log n)^{2d+3};$$

- *for density estimation at* $t = t_0$:

$$\mathbb{E}\mathfrak{H}^2(q_{G^*}^{(t_0)}||q_{\widehat{G}_n}^{(t_0)}) \leq C'_{d,M,(\overline{\sigma},\underline{\sigma}),(\underline{g},\overline{g})} \frac{1}{n}(\log n)^{2d+3};$$

- *for the deconvolution risk,*

$$\mathbb{E}W_2^2 \left( \frac{1}{n}\sum_{i=1}^n (\Sigma_i)^{-1/2}\#G^*, \frac{1}{n}\sum_{i=1}^n (\Sigma_i)^{-1/2}\#\widehat{G}_n \right) \lesssim \frac{1}{\log n}.$$

### B.2  Control of Fisher risk

Our arguments are built upon the "random-$X$" regime with fixed design model (i.e., sample size $n$, latent prior $G^*$, and covariances $\{\Sigma_i\}_{i=1}^n$ are fixed). Conditioned on the fixed design, we can resample the dataset $\mathbf{X}_n = \{X_i\}_{i=1}^n$ from the stochastic processes $\{G^* * \mathcal{N}(0, \Sigma_i)\}_{i=1}^n$. This view is similar to, e.g., Pan et al. [2020] where the observed data as treated random realizations from a fixed design model; see also Rosset and Tibshirani [2020] for discussions on random versus fixed-$X$ statistical viewpoints.

We recall the following definitions of the rescaled mixture laws and its empirical estimator:

$$q_G^{(t)} := \frac{1}{n} \sum_{i=1}^{n} f_{G_{i,t}, I_d}, \quad q_n^{(t)} = \frac{1}{n} \sum_{i=1}^{n} K_t(\cdot \mid i);$$

$$\mathbb{E}_{\mathbf{X}_n}[q_n^{(t)}] = q_G^{(t)},$$

where $G_{i,t} := (\Sigma_i^{(t)})^{-1/2} \# G$ denotes the pushforward measure under linear map induced by the covariance structure $\Sigma_i^{(t)}$ at time $t$, and the rescaled transition kernel that perturbs the $i$-th data pair $(X_i, \Sigma_i)$ is:

$$K_t(\cdot \mid i) = \mathcal{N}\left([\alpha_i(t)\Sigma_i]^{-1/2} X_i, (1 - \alpha_i(t)^{-1}) I_d\right).$$

We abbreviate the score function for $q_G^{(t)}$ and $q_n^{(t)}$ respectively as:

$$s_G^{(t)} := \nabla \log q_G^{(t)}, \qquad s_n^{(t)} := \nabla \log q_n^{(t)}.$$

By definition, the cumulated Fisher divergence $\mathbb{E}\overline{\mathfrak{F}}_{[t_0,T]}(q_{G^*}^{(t)} \| q_G^{(t)})$ between $G^*$ and an arbitrary deterministic measure $G$ is naturally related to the population score matching risk defined by:

$$\mathcal{J}_*(G) := \frac{1}{n} \sum_{i=1}^{n} \int_{t_0}^{T} g_i^2(t) \mathbb{E}_{x \sim f_{G_{i,t}^*, I_d}} \left[\left\| s_G^{(t)}(x) - s_{G^*}^{(t)}(x) \right\|_{\Sigma_i}^2\right] dt.$$

It is clear that the Fisher risk can be upper bounded by $\mathcal{J}_*$ via rescaling of diffusion and noise coefficients:

$$\overline{\mathfrak{F}}_{[t_0,T]}(q_{G^*}^{(t)} \| q_G^{(t)}) \leq \sup_{i \in [n], t \in [t_0, T]} \{[g_i^2(t) \lambda_{\min}(\Sigma_i)]^{-1}\} \mathcal{J}_*(G).$$

Notice however that $\mathcal{J}_*(G)$ needs not to be the expectation of the finite-$n$ empirical risk defined in eq.(11), i.e., $\mathbb{E}_{\mathbf{X}_n} \mathcal{J}_n(G) \neq \mathcal{J}_*(G)$ in general. This is due to that (1) the score estimator is a biased ratio estimator (though the density $q_{G^*}$ and its gradient is unbiasedly estimated): $\mathbb{E}_{\mathbf{X}_n} s_n \neq s_{G^*}$, and (2) both the integrand functional $s_n$ and the sampling law $K_t$ are $\mathbf{X}_n$-dependent thus they cannot be trivially decoupled.

Thus, we further define a decoupled risk with respect to population-level score $s_{G^*}$ and data-dependent sampling law $K_i(\cdot|X_i)$:

$$\widehat{\mathcal{J}}_*(G) := \frac{1}{n} \sum_{i=1}^{n} \int_{t_0}^{T} g_i^2(t) \mathbb{E}_{x \sim K_t(\cdot|i)} \left[\left\| s_G^{(t)}(x) - s_{G^*}^{(t)}(x) \right\|_{\Sigma_i}^2\right] dt.$$

Then it is clear that $\mathbb{E}_{\mathbf{X}_n} \widehat{\mathcal{J}}_*(G) = \mathcal{J}_*(G)$ for every deterministic $G$, because $\mathbb{E}[K_t(\cdot|i)] \equiv f_{G_{i,t}^*, I_d}$.

For the random estimator $\widehat{G}_n$, the optimality condition of $\mathcal{J}_n(\widehat{G}_n)$ implies a basic inequality that allows us to link $\mathcal{J}_*$ and $\mathcal{J}_n$ through $\widehat{\mathcal{J}}_*$, as we detail below.

**Lemma 1** (Basic inequality). *Every optimal solution $\widehat{G}_n$ to (11) satisfies:*

$$\widehat{\mathcal{J}}_*(\widehat{G}_n) \leq 4 \mathcal{J}_n(G^*). \tag{15}$$

*Proof of Lemma 1.* We start with the following definition that presents a joint view over the tuple $(i, t, x)$ to ease the integral representation in $\mathcal{J}_n$.

**Definition 1** (Joint law over index, time and sample). *Fix the data $\{(X_i, \Sigma_i)\}_{i=1}^{n}$ and set*

$$Z = \sum_{i=1}^{n} \int_{t_0}^{T} g_i^2(t) \, dt.$$

*Define the probability measure $\overline{\mathbb{P}}_n$ on $\{1, \dots, n\} \times [t_0, T] \times \mathbb{R}^d$ by*

$$\overline{\mathbb{P}}_n\{i = k, t \in dt, x \in dx\} = \frac{g_k^2(t)}{n Z} \varphi_{(1 - \alpha_k(t)^{-1}) I_d}(x - [\alpha_k(t)\Sigma_k]^{-1/2} X_k) \, dt \, dx.$$

*Equivalently:*

$$\begin{cases} \Pr\{i = k\} = \dfrac{1}{n}, \\ \left(t \mid i = k\right) \text{ has density } \dfrac{g_k^2(t)}{\int_{t_0}^T g_k^2(u)\,du}, \\ \left(x \mid i = k,\ t\right) \sim K_t(\cdot|k). \end{cases}$$

With the above definitions, we first note that we can rewrite the objective (11) by:

$$\mathcal{J}_n(G) = \frac{1}{n} \sum_{i=1}^n \int_{t_0}^T g_i^2(t)\, \mathbb{E}_{x \sim K_t(\cdot|i)} \big\| s_G^{(t)}(x) - s_n^{(t)}(x) \big\|_{\Sigma_i}^2 dt$$

$$= Z\, \mathbb{E}_{(i,t,x) \sim \overline{\mathbb{P}}_n} \big\| s_G^{(t)}(x) - s_n^{(t)}(x) \big\|_{\Sigma_i}^2.$$

By the optimality condition, provided $G \in \mathcal{P}(\mathbb{R}^d)$ is a minimizer of (11) and $G^*$ is feasible, we have $\mathcal{J}_n(G) \le \mathcal{J}_n(G^*)$. Using the quadratic expansion:

$$\|a - c\|_\Sigma^2 - \|b - c\|_\Sigma^2 = \|a - b\|_\Sigma^2 - 2(b - c)^\top \Sigma(a - b) \le 0,$$

where we let $a = s_G^{(t)}$, $b = s_{G^*}^{(t)}$, and $c = s_n^{(t)}$, we obtain:

$$\mathbb{E}_{(i,t,x) \sim \overline{\mathbb{P}}_n} \big\| s_G^{(t)} - s_{G^*}^{(t)} \big\|_{\Sigma_i}^2 \le 2 \mathbb{E}_{(i,t,x) \sim \overline{\mathbb{P}}_n} \left[ (s_{G^*}^{(t)} - s_n^{(t)})^\top \Sigma_i (s_{G^*}^{(t)} - s_G^{(t)}) \right]. \tag{16}$$

By Cauchy-Schwarz inequality applied to the inner product term (16) and rearranging terms,

$$\mathbb{E}_{(i,t,x) \sim \overline{\mathbb{P}}_n} \left[ \big\| s_G^{(t)}(x) - s_{G^*}^{(t)}(x) \big\|_{\Sigma_i}^2 \right] \le 4\, \mathbb{E}_{(i,t,x) \sim \overline{\mathbb{P}}_n} \left[ \big\| s_{G^*}^{(t)}(x) - s_n^{(t)}(x) \big\|_{\Sigma_i}^2 \right],$$

which recovers the claim by rewriting the integrals back in $\mathcal{J}_n$ and $\widehat{\mathcal{J}}_*$. $\qquad\square$

By the basic inequality eq. (15), we can now relate the empirical risk to the population risk as follows:

$$\mathcal{J}_*(\widehat{G}_n) = [\mathcal{J}_*(\widehat{G}_n) - \widehat{\mathcal{J}}_*(\widehat{G}_n)] + \widehat{\mathcal{J}}_*(\widehat{G}_n). \tag{17}$$

The first term is the uniform deviation between the true and empirical $\mathcal{J}_*$ over the feasible set of $G$, and the second term is at most four times the score empirical regret $\mathcal{J}_n(G^*)$ conditioned on the observed data $\mathbf{X}_n$. Below we control the two terms in expectation respectively.

### B.2.1 Control of $\mathcal{J}_n(G^*)$

We first have:

$$\mathcal{J}_n(G^*) = \int_{t_0}^T \frac{1}{n} \sum_{i=1}^n g_i^2(t) \mathbb{E}_{x \sim K_t(\cdot|i)} \left[ \big\| s_{G^*}^{(t)}(x) - s_n^{(t)}(x) \big\|_{\Sigma_i}^2 \right] dt$$

$$\le \int_{t_0}^T \max_{i \in [n]} \{ g_i^2(t) \lambda_{\max}(\Sigma_i) \} \mathbb{E}_{x \sim q_n^{(t)}} \left[ \big\| s_{G^*}^{(t)}(x) - s_n^{(t)}(x) \big\|_2^2 \right] dt.$$

To control this term, we integrate and generalize tools from Wibisono et al. [2024], Saha and Guntuboyina [2020] to prove our next lemma.

**Lemma 2** (Control of score empirical regret). *For each $t \in [t_0, T]$ where $(t_0, T)$ are specified in Theorem 3,*

$$\mathbb{E}_{\mathbf{X}_n} \left[ \mathbb{E}_{x \sim q_n^{(t)}} \big\| s_{G^*}^{(t)}(x) - s_n^{(t)}(x) \big\|_2^2 \right] \le C_{d, M, (\overline{\sigma}, \underline{\sigma})} \frac{1}{n} (\log n)^{d+3}. \tag{18}$$

We now build up the necessary toolbox to prove the above lemma. Following Jiang and Zhang [2009], Saha and Guntuboyina [2020], it is beneficial to consider the following regularized score rule. Given an arbitrary density $\rho$, define its associated regularized score function with parameter $\epsilon > 0$ as:

$$s^\epsilon := \frac{\nabla \rho}{\rho \vee \epsilon},$$

where $\rho \vee \epsilon$ denotes $\max\{\rho, \epsilon\}$. For a fixed $t$, when it is clear in context, we now drop the index $(t)$ in the score. Expand the empirical regret via regularization:

$$\mathbb{E}_{q_n}\left[\|s_{G^*} - s_n\|^2\right] \leq 3\mathbb{E}_{q_n}\left[\|s_{G^*} - s_{G^*}^\epsilon\|^2 + \|s_{G^*}^\epsilon - s_n^\epsilon\|^2 + \|s_n - s_n^\epsilon\|^2\right].$$

As useful in the following proof, we first define a general class of heteroscedastic mixtures, in which both the base measures $G$ and the Gaussian smoothing kernels can be heteroscedastic across components. We then show that their associated regularized score function is uniformly bounded. The definition will be useful to characterize the behavior of the empirical score $s_n$, in which each component is a different Dirac measure $\delta_{(\alpha_i \Sigma_i)^{-1/2} X_i}$ smoothed by a personalized transition kernel $\mathcal{N}(0, K_i)$ with $K_i = (1 - \alpha_i^{-1}) I_d$.

**Definition 2** (General heteroscedastic mixtures). *Let $\{K_i\}_{i=1}^n$ be positive definite covariance matrices such that $\underline{k} I_d \preceq K_i \preceq \overline{k} I_d$. Let $G_{[n]} := \{G_i\}_{i=1}^n$ be a collection of probability measures on $\mathbb{R}^d$. Define the heteroscedastic mixture $p_{G_{[n]}, h} = \frac{1}{n} \sum_{i=1}^n G_i * \mathcal{N}(0, hK_i)$ for bandwidth parameter $h > 0$.*

**Lemma 3** (Uniform bounding of regularized scores). *Fix a mixture density defined in Definition 2. For every $x \in \mathbb{R}^d$,*

$$\frac{\|\nabla p_{G_{[n]}, h}(x)\|_2}{p_{G_{[n]}, h}(x) \vee \epsilon} \leq \sqrt{\frac{1}{h\underline{k}} \log \frac{(2\pi h\underline{k})^{-d}}{\epsilon^2}},$$

*for $0 < \epsilon \leq (2\pi h\underline{k})^{-d/2} e^{-1/2}$.*

*Proof.* For each component $p_i := G_i * \mathcal{N}(0, hK_i)$ in the mixture, consider the transformation such that if $X \sim p_i$, let $\widetilde{X}_i := (hK_i)^{-1/2} X_i$ and let $\widetilde{p}_i = \widetilde{G}_i * \mathcal{N}(0, I_d)$ be the transformed measure governing $\widetilde{X}_i$. Then we have the following equations:

$$p_i(x) = |hK_i|^{-1/2} \widetilde{p}_i\left((hK_i)^{-1/2} x\right),$$

$$\nabla p_i(x) = |hK_i|^{-1/2} (hK_i)^{-1/2} \nabla \widetilde{p}_i\left((hK_i)^{-1/2} x\right).$$

For $\widetilde{p}_i$ being convolution of measure with standard Gaussian kernel, Saha and Guntuboyina [2020, Lemma F.1] directly applies:

$$\frac{\|\nabla \widetilde{p}_i\|}{\widetilde{p}_i} \leq \sqrt{\log \frac{(2\pi)^{-d}}{\widetilde{p}_i^2}}$$

$$\implies \frac{\|\nabla p_i\|}{p_i} = \frac{\|(hK_i)^{-1/2} \nabla \widetilde{p}_i\|}{\widetilde{p}_i} \leq \sqrt{\frac{1}{h\underline{k}} \log \frac{(2\pi)^{-d}}{\widetilde{p}_i^2}} \leq \sqrt{\frac{1}{h\underline{k}} \log \frac{(2\pi h\underline{k})^{-d}}{p_i^2}}.$$

We observe that the function $x\sqrt{\log \beta x^{-2}}$ is concave over $x \in (0, \sqrt{\beta})$, thus by Jensen's inequality:

$$\sum_{i=1}^n \|\nabla p_i\| \leq n p_{G_{[n]}, h} \sqrt{\frac{1}{h\underline{k}} \log \frac{(2\pi h\underline{k})^{-d}}{p_{G_{[n]}, h}^2}},$$

and

$$\frac{\|\nabla p_{G[n], h}\|}{p_{G[n], h}} \leq \frac{\sum_i \|\nabla p_i\|}{n p_{G_{[n]}, h}} \leq \sqrt{\frac{1}{h\underline{k}} \log \frac{(2\pi h\underline{k})^{-d}}{p_{G_{[n]}, h}^2}}.$$

The claim then follows from a same argument as in proof of (F.2) in Lemma F.1 of Saha and Guntuboyina [2020] to relate the regularized rule with the unregularized bound derived above under the two cases $p_{G[n], h} \leq \epsilon$ and $p_{G[n], h} > \epsilon$. $\qquad\square$

We now turn our attention to tools for bounding the discrepancy between regularized scores:

$$\mathbb{E}_{q_n}[\|s_{G^*}^\epsilon - s_n^\epsilon\|^2],$$

via their corresponding Hellinger distance. We rely on specializing Theorem E.1 of Saha and Guntuboyina [2020] and Lemma 1 of Wibisono et al. [2024] to the following form that is closely related to our setting with heteroscedastic mixture.

**Lemma 4** (Bayes discrepancy of regularized scores). *For two arbitrary $n$-collections of probability measures $G_{[n]}$ and $H_{[n]}$ on $\mathbb{R}^d$. Let the mixtures $p_{G_{[n]},h}$ and $p_{H_{[n]},h}$ be defined as in Definition 2, then*

$$\mathbb{E}_{p_{G_{[n]},h}}\left[\left\|s^{\epsilon}_{G_{[n]},h} - s^{\epsilon}_{H_{[n]},h}\right\|_2^2\right]$$
$$\leq \frac{Cd}{h\underline{k}}\max\left\{\left(\log\frac{(2\pi h\underline{k})^{-d/2}}{\epsilon}\right)^3, |\log\mathfrak{H}(p_{G_{[n]},h}, p_{H_{[n]},h})|\right\}\mathfrak{H}^2(p_{G_{[n]},h}, p_{H_{[n]},h}),$$

*if $h\underline{k} \leq 1$ and $0 < \epsilon \leq (2\pi h\underline{k})^{-d/2}e^{-1/2}$.*

*Proof of Lemma 4.* Examining the proof of Saha and Guntuboyina [2020, Theorem E.1], we have:

$$\mathbb{E}_{p_{G_{[n]},h}}\left[\left\|s^{\epsilon}_{G_{[n]},h} - s^{\epsilon}_{H_{[n]},h}\right\|^2\right]^{1/2} \leq T_1 + T_2,$$

with

$$\frac{1}{4}T_1^2 \leq 2\mathfrak{H}^2(p_{G_{[n]},h}, p_{H_{[n]},h})\Upsilon^2, \quad \text{and} \quad \frac{1}{4}T_2^2 \leq \sum_{i=1}^d \Delta_{i,1}^2,$$

where

$$\Upsilon^2 := \sup_{x\in\mathbb{R}^d, \circ\in\{G_{[n]}, H_{[n]}\}}\left\{\frac{\|\nabla p_{\circ,h}(x)\|}{p_{\circ,h}(x)\vee\epsilon}\right\}^2, \quad \Delta_{i,k}^2 := \int\frac{(\partial^k/\partial x_i^k(p_{G_{[n]},h} - p_{H_{[n]},h})^2}{p_{G_{[n]},h}\vee\epsilon + p_{H_{[n]},h}\vee\epsilon}.$$

With integration by parts, it is shown there that $\Delta$'s satisfy a recursion that for $k \geq 1$,

$$\Delta_{i,k}^2 \leq 2\Upsilon\Delta_{i,k-1}\Delta_{i,k} + \Delta_{i,k-1}\Delta_{i,k+1}.$$

With generalized Lemma F.2 of Saha and Guntuboyina [2020] and some work around the recursion, we can deduce that for every $a \geq \sqrt{(2k-1)/(h\underline{k})}$:

$$\Delta_{i,k}^2 \leq \frac{2(2\pi h\underline{k})^{-d/2}}{\epsilon}\left[a^{2k}\mathfrak{H}^2(p_{G_{[n]},h}, p_{H_{[n]},h}) + \sqrt{\frac{2}{\pi}}(a^2/h\underline{k})^{(2k-1)/2}e^{-h\underline{k}a^2}\right],$$

and further with $k = k_0 + 1 \geq \frac{h\underline{k}}{2}\Upsilon^2$ and taking $a\sqrt{h\underline{k}} = \max\left\{\sqrt{2k_0 + 1}, \sqrt{2|\log\mathfrak{H}(p_{n.h}, p_{*,h})|}\right\}$, we have

$$e^{-h\underline{k}a^2} \leq \mathfrak{H}^2(p_{G_{[n]},h}, p_{H_{[n]},h}),$$

and

$$\Delta_{i,1} \leq C\max\left\{2k_0\Upsilon, e(1 + \sqrt{h\underline{k}})a\right\}\mathfrak{H}(p_{n,h}, p_{*,h}).$$

Plugging in the uniform upper bound on $\Upsilon$ by Lemma 3 and some basic algebra around the choice of $k_0$ and merging of terms yield the desired bound. Because the last algebraic steps exactly follow from Saha and Guntuboyina [2020, Proof of Theorem E.1], we refer our readers to their work for more details. $\square$

The above result implies that we can further reduce the discrepancy between regularized scores to bounding the Hellinger distance between $q_{G^*}$ and its empirical estimate $q_n$. We recall the following result from Wibisono et al. [2024] that controls the Hellinger convergence rate of smoothed empirical distributions with the vanilla radial Gaussian kernel.

**Lemma 5** ([Wibisono et al., 2024, Lemmas 2 & 3]). *Let $\rho$ be an $\alpha$-subgaussian measure on $\mathbb{R}^d$ whose score $s(x) = \nabla\log\rho(x)$ is $L$-Lipschitz. Let $\rho_h = \rho * \varphi_h$ where $\varphi_h$ is the density of $\mathcal{N}(0, hI_d)$. Then for every $x \in \mathbb{R}^d$,*

$$\frac{\rho(x)}{\rho_h(x)} \leq \exp\left(dLh/2\right). \tag{19}$$

*Let $\rho_n$ be its size-$n$ empirical measure and $\rho_{n,h} = \rho_n * \mathcal{N}(0, hI_d)$. Then provided $h \leq \min\{\frac{1}{4L}, \alpha^2\}$,*

$$\mathbb{E}\left[\mathfrak{H}^2(\rho_{n,h}, \rho_h)\right] \leq \frac{1}{n}\left(\frac{C\alpha^2\log n}{h}\right)^{d/2} + \frac{4d}{n},$$

*where $C > 0$ is a universal constant.*

We extend the above lemma for empirical heteroscedastic kernel densities.

**Corollary 1** (Hellinger convergence of heteroscedastically smoothed empirical measures). *Let $\{Y_i\}_{i=1}^n$ be independent random variables, each drawn from an $\alpha$-subgaussian, $L$-log-smooth measure $p_i$ on $\mathbb{R}^d$. Consider the heteroscedastic mixture $p_n$ with density: $p_{n,h}(x) := \frac{1}{n}\sum_{i=1}^n \varphi_{h_i I_d}(x - Y_i)$ and denote its population mixture $p_{*,h} = \mathbb{E}[p_{n,h}] = \frac{1}{n}\sum_{i=1}^n p_i * \mathcal{N}(0, hK_i)$. Assume each $\underline{k}I_d \preceq K_i \preceq \overline{k}I_d$, and $h\overline{k} \le \min\{\alpha^2, 1/2L\}$. Then,*

$$\mathbb{E}\left[\mathfrak{H}^2(p_{n,h}, p_{*,h})\right] \le \frac{C_{\underline{k},\overline{k},\alpha}}{n}\left(\frac{\log n}{h}\right)^{d/2} + \frac{4d}{n}, \tag{20}$$

*for a constant $C$ that depends on $\alpha, \overline{k}$.*

*Proof of Corollary 1.* By proof of Lemma 2 in Wibisono et al. [2024], let $B \subset \mathbb{R}^d$,

$$\mathbb{E}\left[\mathfrak{H}^2(p_{n,h}, p_{*,h})\right] \le \int_B \frac{\mathbb{E}[(p_{*,h}(x) - p_{n,h}(x))^2]}{p_{*,h}(x)}dx + 2\int_{B^c} p_{*,h}$$

Fix each $x$, then for each $i$-th component of the empirical mixture $\varphi_{hK_i}(x - Y_i)$, by their trick:

$$\mathbb{E}\varphi_{hK_i}(x - Y_i)^2 = |4\pi hK_i|^{-1/2}\mathbb{E}\varphi_{hK_i/2}(x - Y_i).$$

Thus

$$\text{Var}[p_{n,h}(x)] = \frac{1}{n}\sum_{i=1}^n \text{Var}[\varphi_{hK_i}(x - Y_i)]$$

$$= \frac{1}{n}\sum_{i=1}^n \left(|4\pi hK_i|^{-1/2}\mathbb{E}[\varphi_{hK_i/2}(x - Y_i)] - \mathbb{E}[\varphi_{hK_i}(x - Y_i)]^2\right) \le \frac{(4\pi h\underline{k})^{-d/2}}{n}p_{*,h/2}(x).$$

Combining with above,

$$\mathbb{E}\left[\mathfrak{H}^2(p_{n,h}, p_{*,h})\right] \le \frac{(4\pi h\underline{k})^{-d/2}}{n}\int_B \frac{p_{*,h/2}(x)}{p_{*,h}(x)}dx + 2\int_{B^c} p_{*,h}.$$

Notice that $p_{*,h}$ is subgaussian with parameter at most $\sigma = \sqrt{\alpha^2 + h\overline{k}}$. By letting $B$ be the box region with center $\mathbb{E}_{p_*}[\overline{Y}]$ and radius $\sigma\sqrt{\log n}$, standard concentration inequalities for multivariate subgaussian variables yield $\int_{B^c} p_{*,h} \le \frac{2d}{n}$, and for the first term we can bound:

$$\int_B \frac{p_{*,h/2}(x)}{p_{*,h}(x)}dx \le \text{Vol}(B)\sup_x \frac{p_{*,h/2}(x)}{p_{*,h}(x)} = (4\sigma^2 \log n)^{d/2}\sup_x \frac{p_{*,h/2}(x)}{p_{*,h}(x)},$$

where the last term can be bounded by applying an immediate extension from (19) and Lemma 6 of Wibisono et al. [2024]. For each component in the mixture, consider the rotation $\widetilde{p}_{i,h} = K_i^{-1/2}\#p_i * \mathcal{N}(0, hI_d)$. Then notice $\nabla \log \widetilde{p}_{i,h} = K_i^{1/2}\nabla \log p_{i,h}$. For all $h > 0$ and $x \in \mathbb{R}^d$, let $\widetilde{x} = K_i^{-1}x$ such that:

$$\log \frac{p_{i,h}}{p_{i,2h}}(x) = \log \frac{\widetilde{p}_{i,h}}{\widetilde{p}_{i,2h}}(K_i^{-1/2}x)$$

$$\le \mathbb{E}_{Z\sim\mathcal{N}(0,I_d)}[\log \widetilde{p}_{i,h}(\widetilde{x}) - \log \widetilde{p}_{i,h}(\widetilde{x} - \sqrt{h}Z)]$$

$$= \mathbb{E}_{Z\sim\mathcal{N}(0,I_d)}\int_{-\sqrt{h}}^0 \langle -Z, \nabla \log \widetilde{p}_{i,h}(\widetilde{x} - uZ)\rangle du$$

$$\le \mathbb{E}\int_{-\sqrt{h}}^0 \|Z\| \cdot \|\nabla \log \widetilde{p}_{i,h}(\widetilde{x} - uZ) - \nabla \log \widetilde{p}_{i,h}(\widetilde{x})\|du \le \overline{k}Ldh,$$

which implies $p_{*,h}(x)/p_{*,2h}(x) \le \exp(\overline{k}Ldh)$.

Combining the results, we obtain:

$$\mathbb{E}\left[\mathfrak{H}^2(p_{n,h}, p_{*,h})\right] \le \frac{((\frac{4\alpha^2}{n} + h\overline{k})\log n/4\pi h\underline{k})^{d/2}}{n} + \frac{4d}{n} \le \frac{C_d}{n}\left(\frac{4\alpha^2/h + \overline{k}}{\underline{k}}\log n\right)^{d/2} + \frac{4d}{n}.$$

$\square$

Lastly, we study the discrepancy due to regularizing $q_n$ and $q_{G^*}$, which can be controlled via adapting results from Saha and Guntuboyina [2020]. Specifically, we recall the following result:

**Lemma 6** (Saha and Guntuboyina [2020, Lemma 4.3]). *For a probability measure $G \in \mathcal{P}(\mathbb{R}^d)$ and let $\epsilon \in (0, (2\pi)^{-d/2}e^{-1/2}]$. Let $L(\epsilon) := \sqrt{-\log((2\pi)^d\epsilon^2)}$. Then for every compact set $S \subseteq \mathbb{R}^d$,*

$$\Delta(G, \epsilon) = \int \left(1 - \frac{f_{G,I_d}}{f_{G,I_d} \vee \epsilon}\right)^2 \frac{\|\nabla f_{G,I_d}\|_2^2}{f_{G,I_d}} \leq C_d N(4/L(\epsilon), S)L^d(\epsilon)\epsilon + dG(S^c),$$

*where $N(\delta, S)$ is the regular Euclidean $\delta$-covering of set $S$.*

Intuitively, the score and its regularized counterpart differ only in the tails.

**Corollary 2.** *For an arbitrary $n$-mixture of probability measures $G_{[n]} = \frac{1}{n}\sum_{i=1}^n G_i \in \mathcal{P}(\mathbb{R}^d)$, and let the heteroscedastic mixture $p_{G_{[n]},h}$ defined as in Definition 2. Let $\epsilon \in (0, (2\pi h\overline{k})^{-d/2}e^{-1/2}n^{-1}]$ and define $L(\epsilon) := \sqrt{-\frac{1}{h\underline{k}}\log((2\pi h\overline{k})^d n^2 \epsilon^2)}$. For compact sets $S_1, \dots, S_n \subseteq \mathbb{R}^d$,*

$$\mathbb{E}_{p_{G_{[n]},h}}[\|s_{G_{[n]},h} - s^\epsilon_{G_{[n]},h}\|^2] \leq C_{d,\overline{k},\underline{k}} \frac{1}{n}\sum_{i=1}^n \left(\frac{1}{h}N(4/L(\epsilon), S_i)L^d(\epsilon)\epsilon + \frac{d}{h}G_i(S_i^c)\right).$$

*Proof of Corollary 2.* The strategy to prove this slightly generalized version is similar to proof of Lemma 3, the Proof of Lemma F.7 in Saha and Guntuboyina [2020] and proofs in Section D.2 of Soloff et al. [2025]. Specifically, for every fixed $x$, we can let $\widetilde{G}_i$ denote the distribution of $(hK_i)^{-1/2}\theta_i$ where $\theta_i \sim G_i$ and write $x_i = (hK_i)^{-1/2}x$. This allow us to rescale the terms in the score function:

$$p_{G_{[n]},h}(x) = \frac{1}{n}\sum_{i=1}^n |hK_i|^{-1/2}f_{\widetilde{G}_i, I_d}(x_i),$$

$$\nabla p_{G_{[n]},h}(x) = \frac{1}{n}\sum_{i=1}^n |hK_i|^{-1/2}(hK_i)^{-1/2}\nabla f_{\widetilde{G}_i, I_d}(x_i).$$

By convexity,

$$\frac{\|\nabla p_{G_{[n]},h}(x)\|^2}{p_{G_{[n]},h}(x)} \leq \frac{1}{n}\sum_{i=1}^n \frac{\||hK_i|^{-1/2}(hK_i)^{-1/2}\nabla f_{\widetilde{G}_i, I_d}(x_i)\|^2}{|hK_i|^{-1/2}f_{\widetilde{G}_i, I_d}(x_i)}$$

$$\leq (h\underline{k})^{-(d/2+1)}\frac{1}{n}\sum_{i=1}^n \frac{\|\nabla f_{\widetilde{G}_i, I_d}(x_i)\|^2}{f_{\widetilde{G}_i, I_d}(x_i)},$$

and it is trivial to verify that for every $i \in [n]$:

$$p_{G_{[n]},h}(x) \geq \frac{1}{n}(h\overline{k})^{-d/2}f_{\widetilde{G}_i, I_d}(x_i)$$

$$\implies \left(1 - \frac{p_{G_{[n]},h}(x)}{p_{G_{[n]},h}(x) \vee \epsilon}\right)^2 \leq \left(1 - \frac{f_{\widetilde{G}_i, I_d}(x_i)}{f_{\widetilde{G}_i, I_d}(x_i) \vee n(h\overline{k})^{d/2}\epsilon}\right)^2.$$

Thus,

$$\mathbb{E}_{p_{G_{[n]},h}}[\|s_{G_{[n]},h} - s^\epsilon_{G_{[n]},h}\|^2] = \int \left(1 - \frac{p_{G_{[n]},h}(x)}{p_{G_{[n]},h}(x) \vee \epsilon}\right)^2 \frac{\|\nabla p_{G_{[n]},h}(x)\|^2}{p_{G_{[n]},h}(x)}dx$$

$$\leq C_{d,\underline{k},\overline{k}}\frac{1}{nh}\sum_{i=1}^n \int \left(1 - \frac{f_{\widetilde{G}_i, I_d}(x_i)}{f_{\widetilde{G}_i, I_d}(x_i) \vee n(h\overline{k})^{d/2}\epsilon}\right)^2 \frac{\|\nabla f_{\widetilde{G}_i, I_d}(x_i)\|^2}{f_{\widetilde{G}_i, I_d}(x_i)}dx_i$$

$$= C_{d,\underline{k},\overline{k}}\frac{1}{nh}\sum_{i=1}^n \Delta(\widetilde{G}_i, n(h\overline{k})^{d/2}\epsilon).$$

where $C_{d,\underline{k},\overline{k}} = \overline{k}^{d/2}/\underline{k}^{d/2+1}$. Now, Lemma 6 directly applies to each summand. Provided $\epsilon \in (0, (2\pi h\overline{k})^{-d/2}e^{-1/2}\frac{1}{n}]$, and $L(\epsilon) := \sqrt{-\frac{1}{h\underline{k}}\log((2\pi h\overline{k})^d n^2 \epsilon^2)}$, then for compact sets $S_i$, $i = 1, \ldots, n$,

$$\mathbb{E}_{p_{G_{[n]},h}}[\|s_{G_{[n]},h} - s^{\epsilon}_{G_{[n]},h}\|^2] \leq C_{d,\underline{k},\overline{k}}\frac{1}{nh}\sum_{i=1}^{n}N(4/L(\epsilon), S_i)L^d(\epsilon)\epsilon + dG_i(S_i^c).$$

where we applied the covering number bound and the change-of-variables identity for pushforward measures under linear maps to re-express the terms defined on $S$ and $S^c$.

$\square$

We are now finally ready to integrate the above pieces together to prove Lemma 2.

*Proof of Lemma 2.* Provided

$$q_n^{(t)} = \frac{1}{n}\sum_{i=1}^{n}K_t(\cdot \mid i) = \frac{1}{n}\sum_{i=1}^{n}\delta_{(\alpha_i \Sigma_i)^{-1/2}X_i} * \mathcal{N}(0, (1 - \alpha_i^{-1})I_d)$$

$$q_{G^*}^{(t)} = \frac{1}{n}\sum_{i=1}^{n}f_{G_{i,t}^*, I_d} = \frac{1}{n}\sum_{i=1}^{n}f_{G_{i,t}^*, I_d}\Big|_{t=1} * \mathcal{N}(0, (1 - \alpha_i^{-1})I_d)$$

where each $G_{i,t}^* := (\Sigma_i^{(t)})^{-1/2}\#G^*$ and $\Sigma_i^{(t)} = \alpha_i(t)\Sigma_i = (\int_0^t g_i^2(s)ds)\Sigma_i$. It highlights that $q_n$ and $q_G$ can be viewed as a specialization of the mixtures in Definition 2. We can easily verify that $f_{G_{i,t}^*, I_d}\Big|_{t=1}$ is at most $\sqrt{4M^2/\underline{\sigma} + 1}$-subgaussian and 1-log-smooth. By construction, $1 - \alpha_i^{-1} \in [\frac{1}{2}, 1)$.

By the earlier decomposition,

$$\mathbb{E}_{\mathbf{X}_n}\mathbb{E}_{q_n}\left[\|s_{G^*} - s_n\|^2\right] \leq 3\mathbb{E}_{q_*}\left[\|s_{G^*} - s_{G^*}^{\epsilon}\|^2\right] + 3\mathbb{E}_{\mathbf{X}_n}\mathbb{E}_{q_n|\mathbf{X}_n}\left[\|s_{G^*}^{\epsilon} - s_n^{\epsilon}\|^2 + \|s_n - s_n^{\epsilon}\|^2\right].$$

For the first term, the test function is deterministic with respect to the data, thus we can reduce the expectation to its population quantity. On the other hand, for terms involving $s_n$, it is a random function of the data samples, thus we evaluate first the inner expectation, conditioned on a fixed data.

The first term can be controlled via a direct application of Corollary 2, with $\epsilon = (2\pi)^{-d/2}e^{-1/2}/n$ and the sets $S_i = \{(\Sigma_i^{(t)})^{-1/2}\theta \mid \theta \in [-M, M]^d\}$ so $G_{i,t}^*(S_i) = 1$. Then we have:

$$\mathbb{E}_{q_*}\left[\|s_{G^*} - s_{G^*}^{\epsilon}\|^2\right] \leq C_d[\sup_i N(4, S_i)]\frac{1}{n} \lesssim \left(1 + \frac{M\sqrt{d}}{2(\min_i \alpha_i(t)\underline{\sigma})^{1/2}}\right)^d \frac{1}{n}$$

$$\leq C_{d,M,\underline{\sigma}}\frac{1}{n}.$$

The last term can be controlled similarly. Standard Gaussian concentration yields that: provided $n \geq 2d$, with probability at least $1 - n^{-1}$, $\max\|X_i\|_\infty \leq r_n := M + \overline{\sigma}\sqrt{6\log(n)}$. Now let the compact set in Corollary 2 be $S_i := \{(\Sigma_i^{(t)})^{-1/2}x \mid x \in [-r_n, r_n]^d\}$. Conditioned on the high-probability event, each $\delta_{(\Sigma_i^{(t)})^{-1/2}X_i}$ is compactly supported on $S_i$ and thus satisfies $G_i(S_i^c) = 0$; otherwise $G_i(S_i^c) \leq 1$ is uniformly bounded. Let $\epsilon = (2\pi)^{-d/2}e^{-1/2}/n$ and we obtain:

$$\mathbb{E}_{\mathbf{X}_n}\mathbb{E}_{q_n|\mathbf{X}_n}\left[\|s_n - s_n^{\epsilon}\|^2\right] \lesssim \left(\sup_{i\in[n]}N(4/L(\epsilon), S_i)L(\epsilon)^d + d\right)\frac{1}{n} \lesssim \frac{(\log n)^d}{n}.$$

Lastly for the Bayes discrepancy between regularized scores, by Lemma 4, Corollary 1, and the same strategy to reduce the expected Hellinger distance in Wibisono et al. [2024, Lemma 4],

$$\mathbb{E}_{\mathbf{X}_n}\mathbb{E}_{q_n|\mathbf{X}_n}\left[\|s_{G^*}^{\epsilon} - s_n^{\epsilon}\|^2\right] \leq C_{d,\underline{\sigma}}\mathbb{E}_{\mathbf{X}_n}\left[\max\left\{(\log n)^3, |\log \mathfrak{H}(q_n, q_{G^*})|\right\}\mathfrak{H}^2(q_n, q_{G^*})\right]$$

$$\leq C_{d,M,(\overline{\sigma},\underline{\sigma})}\frac{(\log n)^{d/2+3}}{n}.$$

Combining the terms, we obtain the desired result that for all $t \in [t_0, T]$,

$$\mathbb{E}_{\mathbf{X}_n} \left[ \mathbb{E}_{x \sim q_n^{(t)}} \left\| s_{G^*}^{(t)}(x) - s_n^{(t)}(x) \right\|_2^2 \right] \le C_{d,M,(\bar{\sigma}, \underline{\sigma})} \frac{1}{n} (\log n)^{d+3}.$$

$\square$

## B.2.2 Control of $\mathcal{J}_*(\widehat{G}_n) - \widehat{\mathcal{J}}_*(\widehat{G}_n)$

We control the term $\mathcal{J}_*(\widehat{G}_n) - \widehat{\mathcal{J}}_*(\widehat{G}_n)$ in (17) via empirical process theory together with the regularization trick introduced above. In particular, we crucially rely on local Rademacher complexity analysis following Bartlett et al. [2005], Lei et al. [2016]. Localization also appears essential to obtain sharp learning rate in empirical Bayes guarantees [Saha and Guntuboyina, 2020, Soloff et al., 2025, Stein, 1981]. Under the uniform bounding assumption, it is without loss of generality to let $g_i = g$ for all $i \in [n]$ to ease the notations.

Denote the pointwise loss $\psi$ with:

$$\psi_{i,t}(x; G) := \left\| s_G^{(t)}(x) - s_{G^*}^{(t)}(x) \right\|_{\Sigma_i}^2,$$

and let the regularized point-wise discrepancy $\psi^\epsilon$, where we write the shorthand notation $s_{G,t}^\epsilon := \nabla q_G^{(t)} / (q_G^{(t)} \vee \epsilon)$:

$$\psi_{i,t}^\epsilon(x; G) := \| s_{G,t}^\epsilon(x) - s_{G^*,t}^\epsilon(x) \|_{\Sigma_i}^2.$$

We also let:

$$h_{i,t}(x; G) = \mathbb{E}_{Z \sim \mathcal{N}(0,(1-\alpha_i(t)^{-1})I_d)} \psi_{i,t}(x + Z; G), \quad h_{i,t}^\epsilon(x; G) = \mathbb{E}_{Z \sim \mathcal{N}(0,(1-\alpha_i(t)^{-1})I_d)} \psi_{i,t}^\epsilon(x + Z; G).$$

such that for all fixed $G$, denote $\widetilde{X}_i := (\Sigma_i^{(t)})^{-1/2} X_i$

$$\mathbb{E}[h_{i,t}(\widetilde{X}_i; G)] = \mathbb{E}_{X \sim f_{G_{i,t}^*, I_d}} [\psi_{i,t}(X; G)],$$

$$\widehat{\mathcal{J}}_*(G) = \int g(t)^2 \frac{1}{n} \sum_i h_{i,t}(\widetilde{X}_i; G) dt, \quad \mathcal{J}_*(G) = \mathbb{E}\widehat{\mathcal{J}}_*(G).$$

Consistent with Bartlett et al. [2005], we write for a function $f : \mathcal{X} \to \mathbb{R}$,

$$P_n f = \frac{1}{n} \sum_{i=1}^n f(X_i), \quad Pf = \mathbb{E}f(X), \quad R_n f = \frac{1}{n} \sum_{i=1}^n \sigma_i f(X_i).$$

Here $\sigma_1, \ldots, \sigma_n$ are $n$ independent Rademacher variables. For a function class $\mathcal{F}$, its Rademacher complexity is $\mathbb{E}R_n\mathcal{F} := \mathbb{E}\left[ \sup_{f \in \mathcal{F}} R_n f \right]$.

To facilitate metric entropy toolkit derived in Saha and Guntuboyina [2020], Soloff et al. [2025], we consider the following pseudometric that restricts the localized score discrepancy to a compact set $S$:

$$m_\epsilon^S(G, H) := \max_{i \in [n]} \sup_{x \in S} \left\| s_{G,t}^\epsilon(x) - s_{H,t}^\epsilon(x) \right\|_{\Sigma_i}.$$

It is natural to consider the following compact set: $A := \{ x \in \mathbb{R}^d : q_G^{(t)}(x) \ge \epsilon, \ \forall G \in \mathcal{P}([-M, M]^d) \}$, in which we guarantee $s_G = s_G^\epsilon$ for all measures $G$. Denote

$$(\underline{\sigma}_t, \bar{\sigma}_t) := \left( \min_{i \in [n]} \sigma_{\min}(\Sigma_i^{(t)}), \max_{i \in [n]} \sigma_{\max}(\Sigma_i^{(t)}) \right).$$

It is straightforward to show that a sufficient radial condition that implies $x \in A$ is:

$$\|x\|_2 \le r_\epsilon \le \left[ \sqrt{2 \log((2\pi))^{-d/2}\epsilon^{-1})} - M\sqrt{d/\underline{\sigma}_t} \right]_+.$$

Let $B = \{ x \in \mathbb{R}^d : \|x\|_2 \le r_\epsilon \}$ then $B \subseteq A$. By standard Gaussian concentration, provided

$$\epsilon \le \epsilon_n := (2\pi)^{-d/2} \exp\left( -\frac{1}{2} \left( 2M\sqrt{d/\underline{\sigma}_t} + \sqrt{d} + 2\sqrt{\log n} \right)^2 \right),$$

then

$$\mathbb{P}(X \in A) \geq \mathbb{P}(X \in B) \geq 1 - n^{-2}, \; \forall X \sim f_{G_{i,t}, I_d}, \forall i \in [n]. \tag{21}$$

With the above construction, we define the discrepancy function $\widetilde{h}_{G,t} : [n] \times \mathbb{R}^d \to \mathbb{R}$ that restricts the regularized discrepancy to the high-probability set $A$:

$$\widetilde{h}_{G,t}(i, x) = \mathbb{E}_{Z \sim \mathcal{N}(0, (1-\alpha_i(t)^{-1})I_d)}[\psi_{i,t}^\epsilon(x + Z; G)\mathbf{1}\{x + Z \in A\}].$$

Then for arbitrary fixed $G$, we separate:

$$\mathcal{J}_*(G) - \widehat{\mathcal{J}}_*(G) = \int_{t_0}^T g(t)^2 [P\widetilde{h}_{G,t} - P_n\widetilde{h}_{G,t}]dt$$

$$+ \int_{t_0}^T g(t)^2 \frac{1}{n} \sum_{i=1}^n \left\{ \mathbb{E}[\psi_{i,t}\mathbf{1}\{A^c\}] - \mathbb{E}_{Z_i^{(t)}}\psi_{i,t}(\widetilde{X}_i + Z_i^{(t)})\mathbf{1}\{\widetilde{X}_i + Z_i^{(t)} \in A^c\} \right\} dt$$

$$\leq \int_{t_0}^T g(t)^2 [P\widetilde{h}_{G,t} - P_n\widetilde{h}_{G,t}]dt + \int_{t_0}^T g(t)^2 \frac{1}{n} \sum_{i=1}^n \mathbb{E}[\psi_{i,t}\mathbf{1}\{A^c\}]dt,$$

where $P$ and $P_n$ are defined with respect to the product measures $\frac{1}{n}\sum_{i=1}^n \delta_i \otimes f_{G_{i,t}^*, I_d}\big|_{t=1}$ and $\frac{1}{n}\sum_{i=1}^n \delta_{(i, \widetilde{X}_i)}$ defined over the augmented space $[n] \times \mathbb{R}^d$. We drop the last term since $\psi$'s are nonnegative.

**Control of $[P - P_n]\widetilde{h}$.** We apply the following results that establishes generalization analysis with local Rademacher complexities:

**Lemma 7** (Theorem 3.3 of Bartlett et al. [2005]). *Let $\mathcal{F}$ be a class of functions with range in $[a, b]$ and assume that there are some functional $T : \mathcal{F} \to \mathbb{R}_+$ and some constant $B$ such that for every $f \in \mathcal{F}$, $\mathrm{Var}[f] \leq T[f] \leq B \cdot Pf$. Let $\varphi$ be a sub-root function that satisfies the following properties:*

- $\varphi : [0, \infty) \to [0, \infty)$ *is non-decreasing and* $r \to \varphi(r)/\sqrt{r}$ *is non-increasing for* $r > 0$.

- $\varphi$ *is continuous on* $[0, \infty)$ *and the equation* $\varphi(r) = r$ *has a unique positive solution* $r^*$. *For all* $r >, r \geq r^* \iff r \geq \varphi(r)$.

*If for any $r \geq r^*$, $\varphi$ satisfies,*

$$\varphi(r) \geq B \, \mathbb{E}R_n\{f \in \mathcal{F} : T(f) \leq r\},$$

*then for any $K > 1$, and any $t > 0$, with probability at least $1 - \exp(-t)$,*

$$Pf \leq \frac{K}{K-1}P_n f + c_1 \frac{K}{B}r^* + (c_2(b-a) + c_3 BK))\frac{t}{n}, \quad \forall f \in \mathcal{F}$$

*where $c_0, c_1, c_2$ are numerical constants.*

It is convenient to then apply Lei et al. [2016]'s results that presents the local Rademacher complexity bounds in terms of metric entropy.

**Lemma 8** (Corollary 1 of Lei et al. [2016]). *Let $\mathcal{F}$ be a function class with $\sup_{f \in \mathcal{F}} \|f\|_\infty \leq b$. If $\mathcal{F}$ can be finitely covered with respect to the metric $\| \cdot \|_{L_2(P_n)} := (\int |f|^2 d\mu)^{1/2}$; that is there exists positive numbers $\gamma, \delta, p$ such that $\log N(u, \mathcal{F}, \|\cdot\|_2) := \sup_n \sup_{P_n} \log N(u, \mathcal{F}, \|\cdot\|_{L_2(P_n)}) \leq \delta \log^p(\gamma/u)$ for any $u \in (0, \gamma]$, then for any $r \in (0, \gamma^2]$,*

$$\mathbb{E}R_n\{f \in \mathcal{F} : Pf^2 \leq r\} \leq c_{b,p,\gamma} \left[ \sqrt{\frac{\delta r \log^p(2\gamma n^{1/2})}{n}} + \frac{\delta \log^p(2\gamma n^{1/2})}{n} \right]. \tag{22}$$

In particular, the right hand side of (22) defines a valid sub-root function whose fixed point satisfies $r^* \leq c\delta \log(n)^p/n$ (see Section B.3. of Lei et al. [2016]). Thus, it suffices for us to show the function class of interests satisfies the required covering number bounds.

For every fixed $t$, the (relaxed) function class of interests is $\mathcal{H}_t = \{\widetilde{h}_{G,t} : G \in \mathcal{P}(\mathbb{R}^d)\}$. We have the following:

- For all $h \in \mathcal{H}_t$, there exists $b_t(\epsilon)$ such that $0 \le h \le \psi^\epsilon \le 4 \sup_G \|s_G^\epsilon\|^2 \le b_t^2(\epsilon)$. By Lemma 3, let

$$b_t(\epsilon) = 2\sqrt{\log[(2\pi)^{-d}\epsilon^{-2}]}.$$

- The variance condition is satisfies by letting $T(h) := Ph^2$. Then $\mathrm{Var}[h] \le T[h]$ by definition and $T(h) \le b_t^2(\epsilon)Ph$ by the uniform bounding above.

- The covering number of $\mathcal{H}_t$ satisfies $N(u, \mathcal{H}_t, \|\cdot\|_2) \le N(\frac{u}{2b_t(\epsilon)}, \mathcal{P}(\mathbb{R}^d), m_\epsilon^A)$. For $G, H \in \mathcal{P}(\mathbb{R}^d)$, and $(i, x) \in P_n$,

$$|\widetilde{h}_{G,t} - \widetilde{h}_{H,t}|(i,x) \le 2b_t(\epsilon)\mathbb{E}_Z[|\psi_{i,t}^\epsilon(x + Z; G)^{1/2} - \psi_{i,t}^\epsilon(x + Z; H)^{1/2}| \cdot \mathbf{1}\{x + Z \in A\}].$$

By the reverse triangle inequality that $|\|a - c\| - \|b - c\|| \le \|a - b\|$ and conditioned on $y = x + Z \in A$:

$$
\begin{aligned}
&|\psi_{i,t}^\epsilon(y; G)^{1/2} - \psi_{i,t}^\epsilon(y; H)^{1/2}| \\
&= \left| \|s_{G,t}^\epsilon(y) - s_{G^*,t}^\epsilon(y)\|_{\Sigma_i} - \|s_{H,t}^\epsilon(y) - s_{G^*,t}^\epsilon(y)\|_{\Sigma_i} \right| \\
&\le \|s_{G,t}^\epsilon(y) - s_{H,t}^\epsilon(y)\|_{\Sigma_i} \le m_\epsilon^A(G, H).
\end{aligned}
$$

Thus for every $n$ and $P_n$,

$$
\begin{aligned}
&\|\widetilde{h}_{G,t} - \widetilde{h}_{H,t}\|_{L_2(P_n)} \le 2b_t(\epsilon)m_\epsilon^A(G, H) \\
&\implies \log N(u, \mathcal{H}_t, \|\cdot\|_2) \le \log N\left(\frac{u}{2b_t(\epsilon)}, \mathcal{P}(\mathbb{R}^d), m_\epsilon^A\right).
\end{aligned}
$$

By the same strategy in Soloff et al. [2025], we have:

$$
\begin{aligned}
m_\epsilon^A(G, H) \le &\overline{\sigma}_t^{1/2}(2\epsilon)^{-1}b_t(\epsilon) \sup_{x \in A} |q_G^{(t)}(x) - q_H^{(t)}(x)| \\
&+ \overline{\sigma}_t^{1/2}\epsilon^{-1} \sup_{x \in A} \|\nabla q_G^{(t)}(x) - \nabla q_H^{(t)}(x)\|_2 \\
\le &\overline{\sigma}_t^{1/2}(2\epsilon)^{-1}b_t(\epsilon) \max_{i \in [n]} \sup_{x \in A} |f_{G_{i,t}, I_d}(x) - f_{H_{i,t}, I_d}(x)| \\
&+ \overline{\sigma}_t^{1/2}\epsilon^{-1} \max_{i \in [n]} \sup_{x \in A} \|\nabla f_{G_{i,t}, I_d}(x) - \nabla f_{H_{i,t}, I_d}(x)\|_2.
\end{aligned}
$$

**Lemma 9** (Lemmas 5 and 6 of Soloff et al. [2025]). *Define the semi-norm over $S \subset \mathbb{R}^d$:*

$$
\begin{aligned}
\|f_{G,\circ} - f_{H,\circ}\|_{\infty,S} &:= \max_{i \in [n]} \sup_{x \in S} |f_{G_{i,t}, I_d}(x) - f_{H_{i,t}, I_d}(x)|, \\
\|f_{G,\circ} - f_{H,\circ}\|_{\nabla,S} &:= \max_{i \in [n]} \sup_{x \in S} \|\nabla f_{G_{i,t}, I_d}(x) - \nabla f_{H_{i,t}, I_d}(x)\|.
\end{aligned}
$$

*and let $S^a := \{x \in \mathbb{R}^d : \inf_{y \in S} \|x - y\| \le a\}$. For $\mathbb{F} := \{(f_{G_i, K_i})_{i=1}^n : \{G_i\}_{i=1}^n \in \mathcal{P}(\mathbb{R}^d), \underline{k}I_d \preceq \{K_i\}_{i=1}^n \preceq \overline{k}I_d\}$, there exists positive constants $C_d$ and $c_{d,\overline{k},\underline{k}}$ such that for all compact $S \subset \mathbb{R}^d$, $M > 0$, and sufficiently small $\eta > 0$ such that*

$$\log N(\eta, \mathbb{F}, \|\cdot\|_{\infty,S}) \le C_d N(a, S^a)\left(\log \frac{c_{d,\overline{k},\underline{k}}}{\eta}\right)^{d+1},$$

$$\log N(\eta, \mathbb{F}, \|\cdot\|_{\nabla,S}) \le C_d N(a, S^a)\left(\log \frac{c_{d,\overline{k},\underline{k}}}{\eta}\right)^{d+1}.$$

*where $a = \sqrt{-2\overline{k}\log \frac{\sqrt{\underline{k} \wedge 1}}{5}(2\pi\underline{k})^{d/2}\eta}$.*

Apply Lemma 9 to our metric, by letting

$$u^* = \overline{\sigma}_t^{1/2}(b_t(\epsilon)/2 + 1)\frac{\eta}{\epsilon},$$

provided $u^*$ sufficiently small, we have

$$\log N(u^*, \mathcal{P}(\mathbb{R}^d), m_\epsilon^A) \leq \log N(\eta/2, \mathbb{F}, \|\cdot\|_{\infty,A}) + \log N(\eta/2, \mathbb{F}, \|\cdot\|_{\nabla,A})$$

$$\leq C_d N(a, A^a)(\log \frac{c_d}{\eta})^{d+1}$$

$$\leq C_d a^{-d} \text{Vol}(A^{3a/2})(\log \frac{c_d(b_t(\epsilon)/2+1)}{\epsilon u^*})^{d+1}.$$

where the last step is due to Lemma F.6 of Saha and Guntuboyina [2020]. Since a necessary radial condition for $x \in A$ is given by

$$d(x, [-\frac{M}{\sqrt{\underline{\sigma}_t}}, \frac{M}{\sqrt{\underline{\sigma}_t}}]^d)^2 = \inf_{y \in [-\frac{M}{\sqrt{\underline{\sigma}_t}}, \frac{M}{\sqrt{\underline{\sigma}_t}}]^d} \|x - y\|_2^2 \leq 2\log((2\pi)^{-d/2}\epsilon^{-1}) =: \bar{r}_t^2(\epsilon),$$

we have that:

$$a^{-d}\text{Vol}(A^{3a/2}) \leq a^{-d} \cdot C_d \left(\bar{r}_t(\epsilon) + M\sqrt{d/\underline{\sigma}_t} + \frac{3}{2}a\right)^d$$

$$\lesssim_{d,M,\underline{\sigma}_t} (\bar{r}_t(\epsilon) \vee 1)^d \log \left(\frac{5(2\pi)^{-d/2}}{\eta}\right)^{d/2}.$$

This implies that:

$$\log N\left(\frac{u^*}{2b_t(\epsilon)}, \mathcal{P}(\mathbb{R}^d), m_\epsilon^A\right) \lesssim_{d,M,\underline{\sigma}_t} (\bar{r}_t(\epsilon) \vee 1)^d \left(\log \frac{C'(b_t(\epsilon)/2+1)b_t(\epsilon)}{\epsilon u^*}\right)^{3d/2+1}.$$

By choosing $\epsilon \equiv \epsilon_n = o(n^{-2})$, we have $b_t(\epsilon_n) \asymp \sqrt{\log n}$ and $\bar{r}_t^2(\epsilon_n) \asymp \log n$, provided $n \geq N_{d,M,\overline{\sigma}_t,\underline{\sigma}_t}$ sufficiently large such that terms involving $n$ dominate constant terms. Thus, one can apply Lemma 7 (with $K = 2$) and Lemma 8 (with $\gamma \asymp (b_t(\epsilon)/2+1)b_t(\epsilon)/\epsilon$, $\delta \asymp (\bar{r}_t(\epsilon) \vee 1)^d$, $p = 3d/2+1$) to show that for arbitrary $k \geq 1$ and $G \in \mathcal{P}([-M, M]^d)$, with probability at least $1 - n^{-k}$,

$$P\widetilde{h}_{G,t} - P_n\widetilde{h}_{G,t} \leq P_n\widetilde{h}_{G,t} + c_1\frac{2(\bar{r}_t(\epsilon) \vee 1)^d}{b_t(\epsilon)^2} \cdot \frac{\log(n)^{3d/2+1}}{n} + c_2 b_t(\epsilon)\frac{k\log(n)}{n}$$

$$\implies \int_{t_0}^T g(t)^2[P\widetilde{h}_{\widehat{G}_n,t} - P_n\widetilde{h}_{\widehat{G}_n,t}]dt \leq \widehat{\mathcal{J}}_*(\widehat{G}_n) + O(\frac{(\log n)^{2d}}{n}),$$

since $\widetilde{h}_{G,t}(i, x) \leq h_{i,t}(x; G)$ and the time integral of $P_n h_{i,t}(x; G)$ recovers $\widehat{\mathcal{J}}_*(G)$ by definition.

**Control of tail.** We remain to bound for every $t$ and $i$:

$$\mathbb{E}_{x \sim G_{i,t}^* * \mathcal{N}(0,I_d)}[\psi_{i,t}(x; G)\mathbf{1}\{x \in A^c\}]$$

$$= \mathbb{E}[(\psi_{i,t}(x; G) - \psi_{i,t}^\epsilon(x; G))\mathbf{1}\{x \in A^c\}] + \mathbb{E}[\psi_{i,t}^\epsilon(x; G)\mathbf{1}\{x \in A^c\}].$$

The second term is trivially bounded via uniform bounding $b_t(\epsilon)$ of every regularized score functional $\psi^\epsilon$ and eq.(21) that $\mathbb{P}(A^c) \leq n^{-2}$ provided $\epsilon \equiv \epsilon_n$ chosen above. Thus,

$$\mathbb{E}[\psi_{i,t}^\epsilon(x; G)\mathbf{1}\{x \in A^c\}] \leq b_t^2(\epsilon)n^{-2}. \tag{i}$$

By standard algebraic operations via triangle inequality, let $\delta_1 = \max_{c \in \{a,b\}} \|c - c^\epsilon\|$ and $\delta_2 = \max_{c \in \{a,b\}} \|c^\epsilon\|$,

$$\|a - b\| \leq \|a^\epsilon - b^\epsilon\| + \|a - a^\epsilon\| + \|b - b^\epsilon\| = \|a^\epsilon - b^\epsilon\| + 2\delta_1$$

$$\implies |\|a - b\|^2 - \|a^\epsilon - b^\epsilon\|^2| \leq |\|a - b\| - \|a^\epsilon - b^\epsilon\|| \cdot (\|a - b\| + \|a^\epsilon - b^\epsilon\|)$$

$$\leq 2\delta_1 \cdot (2\|a^\epsilon - b^\epsilon\| + 2\delta_1) \leq 4(\delta_1^2 + 2\delta_1\delta_2).$$

Letting $a = s_G^{(t)}, b = s_{G^*}^{(t)}, a^\epsilon = s_{G,t}^\epsilon, b^\epsilon = s_{G^*,t}^\epsilon$, and applying Cauchy–Schwarz inequality, the first term in above becomes:

$$\mathbb{E}[(\psi_{i,t} - \psi_{i,t}^\epsilon)\mathbf{1}\{A^c\}] = \mathbb{E}[(\|s_G - s_{G^*}\|_{\Sigma_i^{(t)}}^2 - \|s_{G^*}^\epsilon - s_{G^*}^\epsilon\|_{\Sigma_i^{(t)}}^2)\mathbf{1}\{A^c\}]$$

$$\leq \max_{H,H' \in \mathcal{P}([-M,M]^d)} 4\overline{\sigma}_t \mathbb{E}[(\|s_H - s_H^\epsilon\|^2 + 2\|s_H - s_H^\epsilon\| \cdot \|s_{H'}^\epsilon\|)\mathbf{1}\{A^c\}]$$

$$\leq \max_{H \in \mathcal{P}([-M,M]^d)} 4\overline{\sigma}_t \mathbb{E}[(\|s_H - s_H^\epsilon\|^2] + b_t(\epsilon)(\mathbb{E}[\|s_H - s_H^\epsilon\|^2])^{1/2}\mathbb{P}(A^c)^{1/2}$$

$$\leq 4\overline{\sigma}_t \left(\Delta + \frac{b_t(\epsilon)}{n}\Delta^{1/2}\right). \tag{ii}$$

where we again used the uniform bounding . The discrepancy $\Delta := \mathbb{E}[\|s_H - s_H^\epsilon\|^2]$ can be bounded by Corollary 2 with a change of measure. By the Cauchy-Schwarz inequality,

$$\Delta \leq \sqrt{1 + \chi^2(q_H \| q_{G^*})} (\mathbb{E}_{q_H} \|s_H - s_H^\epsilon\|^4)^{1/2},$$

where the fourth moment of the score discrepancy can be analogously controlled as the second moment with only the change of the leading constant and the dependency from $L^d(\epsilon)$ to $L^{d+2}(\epsilon)$. The chi-squared distance $\chi^2(p\|q) = \int \frac{p^2}{q} - 1$ is jointly convex and we have $\chi^2(f_{G_{i,t},I_d} \| f_{H_{i,t},I_d}) \leq C_{d,M,\underline{\sigma},\overline{\sigma}}$ is upper bounded by a constant. Consequently, given $G^*, H$ compactly supported, with the choice of $\epsilon \lesssim n^{-2}$, we have that:

$$\Delta \lesssim \frac{(\log n)^{(d+2)/4}}{n}.$$

Recall $b_t(\epsilon) \asymp \sqrt{\log n}$, so combining (i) + (ii) yields:

$$\int_{t_0}^{T} g(t)^2 \frac{1}{n} \sum_{i=1}^{n} \mathbb{E}[\psi_{i,t} \mathbf{1}\{A^c\}] dt \lesssim \frac{(\log n)^{(d+2)/4}}{n}.$$

Combining the high-probability part and the tail, we can conclude this section by asserting:

$$\mathcal{J}_*(\widehat{G}_n) - \widehat{\mathcal{J}}_*(\widehat{G}_n) \lesssim_{d,M,(\overline{\sigma},\underline{\sigma}),(\underline{g},\overline{g})} \widehat{\mathcal{J}}_*(\widehat{G}_n) + \frac{\text{polylog}(n)}{n}.$$

By further upper bounding $\widehat{\mathcal{J}}_*(\widehat{G}_n)$ by $\mathcal{J}_n(G^*)$ from the basic inequality and applying Lemma 2, we recover the first claim in Theorem 3 on score estimation guarantee that with probability at least $1 - n^{-k}$ for arbitrary $k \geq 1$,

$$\mathbb{E}\mathcal{J}_*(\widehat{G}_n) \lesssim 8\mathbb{E}\mathcal{J}_n(G^*) + \frac{(\log n)^{2d+1/2}}{n} + \frac{\log(n)^{O(1)}}{n} \lesssim \frac{1}{n}(\log n)^{O(d)}.$$

## B.3 Control of Hellinger divergence

Our next result controls the KL divergence between $q_G^{(t_0)}$ and $q_{G^*}^{(t_0)}$ by the Fisher risk controlled above. Without loss of generality, assume $g_i(t) = g(t)$ so $\alpha_i(t) = \alpha(t)$. Recall that we define:

$$q_G^{(t)} = \frac{1}{n} \sum_{i=1}^{n} (\Sigma_i^{(t)})^{-1/2} \# G * \mathcal{N}(0, I_d).$$

Let

$$\widetilde{q}_G^{(t)} = \frac{1}{n} \sum_{i=1}^{n} (\Sigma_i)^{-1/2} \# G * \mathcal{N}(0, \alpha(t)I_d)$$

$$\implies \nabla \log q_G^{(t)}(x) = \frac{1}{\sqrt{\alpha(t)}} \nabla \log \widetilde{q}_G^{(t)}(x/\sqrt{\alpha(t)}).$$

**Lemma 10.** *For every fixed $G \in \mathcal{P}(\mathbb{R}^d)$,*

$$\text{KL}(q_{G^*}^{(t_0)} \| q_G^{(t_0)}) \lesssim \overline{\mathfrak{F}}_{[t_0,T]}(q_{G^*}^{(t_0)} \| q_G^{(t_0)}).$$

*Proof.* Denote the path space by $\Omega = \mathcal{C}([t_0, T]; \mathbb{R}^d)$. Under the latent measure $G$ and the same conditions on $\Sigma_i$ and $g_i$ as in Assumption 1, define the path law $Q_G$ to be the unique solution for the following reverse-time SDE on $[t_0, T]$:

$$dX^{(t)} = -g(t)^2 \nabla \log \widetilde{q}_G^{(t)}(X^{(t)}) \, dt + g(t) \, dB_t, \qquad X^{(t_0)} \sim \widetilde{q}_G^{(t_0)}. \tag{23}$$

Similarly to Song et al. [2021b, Proof of Theorem 1], consider the measurable Markov kernel $K : \Omega \times \mathcal{B}(\mathbb{R}^d) \to [0,1]$ with $K(\omega, x) = \delta_{\omega(t_0)=x}$, then we have:

$$\widetilde{q}_G^{(t_0)} = Q_G K := \int K(\omega, \cdot) Q_G(d\omega)$$

that shows that the marginal mixture law $\widetilde{q}_G$ at $t_0$ is the push-forward of the aggregated path measure $Q_G$ under the kernel $K$. By the data processing inequality,

$$\mathrm{KL}(\widetilde{q}_{G^*}^{(t_0)}||\widetilde{q}_G^{(t_0)}) = \mathrm{KL}(Q_{G^*}K||Q_GK) \leq \mathrm{KL}(Q_{G^*}||Q_G).$$

In addition, the KL divergence is invariant with the linear rescaling:

$$\mathrm{KL}(q_{G^*}^{(t_0)}||q_G^{(t_0)}) = \mathrm{KL}(\widetilde{q}_{G^*}^{(t_0)}||\widetilde{q}_G^{(t_0)}).$$

Since the SDEs associated with $Q_G$ and $Q_{G^*}$ differ only in the drift coefficient, we can apply the Girsanov theorem [Oksendal, 2013] to yield:

$$
\begin{aligned}
\mathrm{KL}(\widetilde{Q}_{G^*} \,\|\, \widetilde{Q}_G) &= \frac{1}{2}\mathbb{E}_{Q_{G^*}}\left[\int_{t_0}^{T} g(t)^2 \left\|\nabla \log \widetilde{q}_{G^*}^{(t)} - \nabla \log \widetilde{q}_G^{(t)}\right\|_2^2 dt\right] \\
&= \frac{1}{2}\int_{t_0}^{T} \frac{g(t)^2}{\alpha(t)}\mathbb{E}_{x \sim q_{G^*}^{(t)}}\left[\left\|\nabla \log q_{G^*}^{(t)} - \nabla \log q_G^{(t)}\right\|_2^2\right]dt \\
&\lesssim \frac{1}{2n}\sum_{i=1}^{n}\int_{t_0}^{T}\mathbb{E}_{x \sim f_{G_{i,t}^*, I_d}}\left[\left\|\nabla \log q_{G^*}^{(t)} - \nabla \log q_G^{(t)}\right\|_2^2\right]dt \\
&\lesssim \frac{1}{2}\overline{\mathfrak{F}}_{[t_0, T]}(q_{G^*}^{(t)}||q_G^{(t)}).
\end{aligned}
$$

$\square$

Plugging in the first claim that bounds $\overline{\mathfrak{F}}_{[t_0, T]}(q_{G^*}^{(t)}||q_G^{(t)})$ in Lemma 10 and use the basic divergence inequality that $\mathfrak{H}^2 \leq KL$ yields the desired second claim.

Lastly, the third claim on the Wasserstein divergence:

$$\mathbb{E}W_2^2\left(\frac{1}{n}\sum_{i=1}^{n}(\Sigma_i)^{-1/2}\#G^*, \frac{1}{n}\sum_{i=1}^{n}(\Sigma_i)^{-1/2}\#\widehat{G}_n\right) \lesssim \frac{1}{\log n}$$

follows from a direct application of Theorem 10 of Soloff et al. [2025] with the upper bound of $\mathfrak{H}^2(\widetilde{q}_G^{(t_0)}, \widetilde{q}_{\widehat{G}_n}^{(t_0)})$ from above.

## B.4 Reduction with Homoscedastic Noise

*Proof of Theorem 1.* When $\Sigma_i = \Sigma$ are all equal, We have:

$$
\begin{aligned}
q_G^t = f_{(\Sigma^{(t)})^{-1/2}\#G, I_d} \implies f_{G,\Sigma^{(t)}}(x) &= |\Sigma^{(t)}|^{-1/2}q_G^t\left((\Sigma^{(t)})^{-1/2}x\right), \\
\nabla \log f_{G,\Sigma^{(t)}}(x) &= (\Sigma^{(t)})^{-1/2}\nabla \log q_G^{(t)}\left((\Sigma^{(t)})^{-1/2}x\right).
\end{aligned}
$$

We have the following equivalence:

$$
\begin{aligned}
\mathfrak{F}(f_{G,\Sigma^{(t)}}\|f_{H,\Sigma^{(t)}}) &\lesssim \lambda_{\min}(\Sigma^{(t)})^{-1}\mathfrak{F}(q_G^t\|q_H^t), \\
\mathrm{KL}(f_{G,\Sigma^{(t)}}\|f_{G,\Sigma^{(t)}}) &= \mathrm{KL}(q_G^t\|q_H^t), \\
W_2^2(G, H) &\leq \sqrt{\lambda_{\max}(\Sigma)}W_2^2(\Sigma^{-1/2}\#G, \Sigma^{-1/2}\#H).
\end{aligned}
$$

This shows that Theorem 1 is a direct corollary of Theorem 3 with only adjustments to the leading constants. $\square$

## C  Implementation Details and Additional Results

**Experiments compute resources.** We fit all our models and performed all experiments on an internal HPC cluster. Jobs were scheduled on GPU-enabled nodes, each using one NVIDIA GeForce GTX 2080 Ti GPU and its accompanying Intel Xeon Skylake 6130 @ 2.1 GHz CPUs with 96 GB of RAM per node. Individual training runs require a maximal of 2 GPU hours and 4 CPU hours, while

the denoising runs require a maximal of 1 GPU hour and 2 CPU hours. The entire experiment takes approximately 120 hours. These estimates do not include preliminary or failed experiments.

**Hyperparameters and architectures.** For our implementations, we consider two variants SDGM-C and SDGM-D, for continuous and discrete latent model, respectively. In the continuous case, the flow uses four repeats of the following block: ActNorm $\rightarrow$ random permutation $\rightarrow$ MaskedPiece-wiseRationalQuadraticAutoregressiveTransform, where each autoregressive transform conditions on all previous dimensions via a 2-layer MADE network (128 hidden features), uses eight linear-tail bins (tail bound 10), enforces minimum bin width/height and derivative of $5 \times 10^{-3}$, leaky ReLU activations, residual blocks, and no dropout or batch norm. The base distribution is a standard normal distribution. In the discrete case, we use a trainable mixture of 2,000 components (initialized from random normals or random draws from the observations) and corresponding randomly initialized weights.

For the baselines, used the official implementation of Dual-ALM provided by the authors of Zhang et al. [2022] at `https://github.com/YangjingZhang/Dual-ALM-for-NPMLE`. For NPMLE-N, we use all available training data samples as the fixed supports. Thus, the latent mixture is given by: $\sum_{j=1}^{n} w_j \delta_{X_j}$ with $w_j$'s trainable. For PEM, we fix a latent mixture of 2,000 components, i.e., $\widehat{G} = \sum_{j=1}^{2000} w_j \delta_{\mu_j}$ with both $\{(w_j, \mu_j)\}_{i=1}^{2000}$ optimizable. We randomly sample points from the dataset as the initial supports and uniformly initialize the weights.

**SDE setting and training objective.** Throughout our experiments, we consider the following family of variance-exploding SDEs:

$$dY_i^{(t)} = g(t)\Sigma_i^{1/2} dB_t, \quad \text{with } Y_i^{(0)} \sim G^*.$$

In particular, we use a shared diffusion coefficient $g(t) = \sqrt{\frac{2}{e^2-1}} e^t$. Under this SDE, the marginal distribution at any time $t \geq 0$ is $Y_i^{(t)} \sim G^* * \mathcal{N}(0, \Sigma_i^{(t)})$, where the effective time-dependent covariance is $\Sigma_i^{(t)} = \frac{e^{2t}-1}{e^2-1}\Sigma_i$. Our observed data $X_i$ correspond to $Y_i^{(1)}$.

The training objective for our SDGM models is based on DSM to match the conditional score of $f_{\widehat{G}, \Sigma_i^{(t)}}$ with $p_{1t}(\cdot|X_i)$. For each data point $(X_i, \Sigma_i)$ in a training batch of size $N_B$:

1. A time $t_i > 1$ is sampled (specifically, $t_i$ is drawn uniformly from $[1 + \epsilon, T_{\max}]$, where we use $\epsilon = 10^{-3}$ and $T_{\max} = 1.5$ in our experiments).

2. A perturbed data point, $X_i'$, is generated by corrupting $X_i$ with additional noise: $X_i' = X_i + Z_i$, where $Z_i \sim \mathcal{N}(0, \Sigma_i^{(t_i)} - \Sigma_i^{(1)})$. This $X_i'$ is a sample from the SDE path at time $t_i$, conditioned on $X_i$ being the state at $t = 1$.

To compute the score of our model's current estimate of the data distribution at time $t_i$, we first represent our learned latent prior, $\widetilde{G}$.

- For SDGM-C, $\widetilde{G}$ is represented by a neural spline flow. We obtain a discrete approximation by drawing $m = 2,000$ samples $\{\mu_j\}_{j=1}^{m}$ from this flow and assigning uniform weights $w_j = 1/m$.

- For SDGM-D, $\widetilde{G} = \sum_{j=1}^{m} w_j \delta_{\mu_j}$, where the support points $\{\mu_j\}_{j=1}^{m}$ and their corresponding logits (which determine weights $w_j$ via softmax) are directly trainable parameters, with $m = 2,000$.

The score of the marginal distribution conditional on $i$-th SDE at time $t_i$ implied by $\widetilde{G}$ and noise covariance $\Sigma_i^{(t_i)}$ is denoted $\nabla \log f_{\widetilde{G}, \Sigma_i^{(t_i)}}(X_i')$. This score is computed as $\sum_{j=1}^{m} p(\mu_j|X_i', \Sigma_i^{(t_i)})(\Sigma_i^{(t_i)})^{-1}(\mu_j - X_i')$, where $f_{\widetilde{G}, \Sigma_i^{(t_i)}}(x) = \sum_{j=1}^{m} w_j \mathcal{N}(x; \mu_j, \Sigma_i^{(t_i)})$ and $p(\mu_j|X_i', \Sigma_i^{(t_i)})$ are the posterior probabilities of the mixture components given the perturbed data $X_i'$. The objective function minimizes the expected squared difference between this model score and the true score of the perturbation kernel, $\nabla_{X_i'} \log p_{1t_i}(X_i'|X_i) = -(\Sigma_i^{(t_i)} - \Sigma_i^{(1)})^{-1} Z_i$. For a batch,

the loss is:

$$\mathcal{L}(\widetilde{G}) = \frac{1}{N_B} \sum_{i=1}^{N_B} g^2(t_i) \left\| \nabla \log f_{\widetilde{G}, \Sigma_i^{(t_i)}}(X_i') - \nabla \log p_{1t_i}(X_i'|X_i) \right\|_{\Sigma_i}^2$$

All our experiments are performed with the importance sampling strategy as in Song et al. [2021b] that further reduce the variance in estimating the time integral by directly sample $t_i$ according to $p(t) \propto g(t)^2$. The optimization is performed using the Adam optimizer. For the experiments presented, we used a learning rate of $5 \times 10^{-5}$ (unless specified otherwise for particular experiments), $\beta_1 = 0.9$, Adam $\epsilon = 10^{-8}$, no weight decay (weight decay = 0.0), and gradient norm clipping at 1.0.

**Denoising.** For denoising, we run the "dopri5" ODE solver implemented in `torchdiffeq` [Chen, 2018] with absolute and relative tolerance fixed at $10^{-4}$ from $t = 0.999$ to $t = 0.001$. We use the same procedure as in the training process to obtain a discrete approximation of the learned latent prior for estimating the score along the reverse-time path.

**More numerical results.** We performed an additional synthetic experiment in which the nontrivial structure in the first two dimensions is discretely supported:

Example 3 (discrete): The first two coordinates of each $\theta_i^*$ are drawn uniformly from the discrete ensembles of three points $\{(0,0), (0,6), (6,0)\}$.

The noises are then drawn with the same procedure as described earlier to produce the observations.

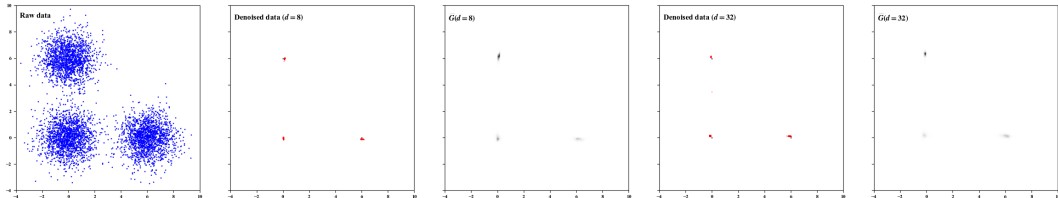

Figure 5: Projected plots of the discrete experiment onto the first two dimensions in dimensions $d \in (8, 16)$ and training sample size $n = 2 \times 10^4$. SDGM-C stays effective in estimating the discrete locations as $d$ grows. Recent NPMLE solver in Zhang et al. [2022] demonstrates significantly degraded performance in such dimensions.

We also performed additional circle experiments with $d$ further increases.

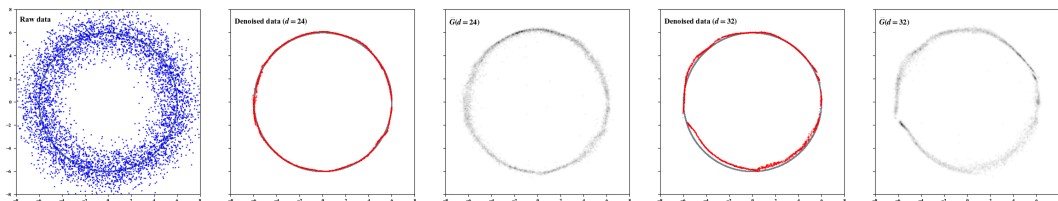

Figure 6: Projected plots of the circle experiment onto the first two dimensions in higher dimensions $d \in (24, 32)$ and training sample size $n = 10^7$. SDGM-C stays relatively stable in recovering the underlying circular structure as $d$ grows to a dimension for which numerical NPMLE solvers struggle.

