# OpenReview forum: "Score-Based Diffusion Modeling for Nonparametric Empirical Bayes in Heteroscedastic Gaussian Mixtures"
_NeurIPS.cc/2025/Conference — NeurIPS 2025 poster_

### Official Review · Reviewer_KpwC · 2025-06-29

**Clarity:** 3
**Significance:** 3
**Originality:** 3
**Rating:** 5
**Confidence:** 3

**Summary:**

The paper proposes a method to integrate nonparametric empirical Bayes with score-based diffusion generative  models. The goal is to estimate an unknown prior distribution from noisy observations. In doing so, the paper addresses both statistical and computational limitations of the classical approach, which combines the nonparametric maximum likelihood estimator (NPMLE) and Tweedie-based posterior denoising. The proposed method generalizes and improves upon existing approaches, which either do not account for heteroscedastic measurement noise or lack theoretical guarantees in dimensions higher than $1$.

The key idea is to reinterpret each noisy observation as a time-1 snapshot from a latent stochastic diffusion process. This perspective naturally induces a denoising score matching objective for explicitly learning the prior distribution $G^*$. The resulting optimization problem is infinite-dimensional and nonconvex. The posterior mean is then estimated by combining this approach with a multi-step Tweedie-type empirical Bayes denoising procedure.

Assuming access to the minimizer of the non-convex optimization problem, the paper provides theoretical guarantees on the statistical performance of the method. Finally, numerical experiments on synthetic and real-world datasets are presented to validate the theory, with particular attention paird to robustness as the dimension of the problem varies.

**Questions:**

- I appreciate a lot how the paper is organized. However, I wonder why the authors do not emphasize more strongly the result regarding the interplay between $n$ and $d$ in the presentation, since dimension plays a role also in the experiments.

- I am confused by the example about the over-shrinkage bias. Isn't the reduced variance simply a consequence of observing $X$, which provides information and hence reduces uncertainty in the posterior estimation?

- How difficult would it be to quantitatively incorporate the optimization error of the nonconvex estimator into the statistical analysis of the method?

**Ethical Concerns:**

["NO or VERY MINOR ethics concerns only"]

**Final Justification:**

As of writing:

- The paper is well-written and provides both a theoretical treatment and numerical experiments that illustrate the theory.
- The authors promised to extend the discussion on the interplay between $n$ and $d$, which I think is a relevant aspect of the paper.

I am still unsure about the assumption concerning the solvability of the infinite-dimensional nonconvex problem. However, after reading the author's rebuttal and the other reviews, I realize that this concern might not represent a major obstacle.

**Limitations:**

See Weaknesses.

**Paper Formatting Concerns:**

I didn't notice any major formatting issues.

**Quality:**

3

**Strengths And Weaknesses:**

**Strenght:**

The paper is well-written and provides both a theoretical treatment and numerical experiments that illustrate the theory.  One interesting feature of the analysis is that the error bounds highlight the interplay between the number of observations and the dimensionality of the problem – something that previous literature didn’t address in this context. Finally, some assumptions are needed to make the analysis clean; these are carefully discussed with reference to prior literature.


**Weakness:**

To my understanding, the paper does not provide strong algorithmic guarantees. To me, this is not a major issue, since the authors are transparent about their assumptions; however, these limitations should be discussed explicitly in a “Limitations” section, which is currently missing. For example, assuming solvability of the nonconvex problem is, I believe (though I am happy to change my mind based on the authors’ response), quite strong, as the optimization can easily get stuck in a local minimum. It would also be interesting to incorporate in the error analysis of Theorem 1-2 the fact that $\widehat{G}$ may not be a good  approximation of the minimizer of the nonconvex objective.

I also noticed a typo on line 182: a repeated “that”.

---

> ### Author Rebuttal · Authors · 2025-07-30
>
> Dear reviewer **KpwC**,
>
> We truly appreciate your recognition of our organization and contributions. Indeed, it is the diffusion reinterpretation integrated with the empirical Bayes G-modeling that jointly and naturally permits both efficient score-based learning of latent measure and the multi-step Tweedie-type denoising procedure. This is not achievable with a classical score-based approach with direct parametrization of the score function. Our results did showcase that we tackled to a satisfactory extent the over-shrinkage bias and scalability challenges of traditional likelihood-based methods.
>
> We also want to thank reviewers for bringing excellent suggestions based upon which we would like to further improve our manuscript. For each question, we intend to both clarify our understanding, and discuss plans to update our manuscript if we have the opportunity to submit a final version. Due to space limits, we may need to refer you to our responses to other reviewers’ related questions. We apologize for such inconvenience and we are happy to follow up with any persisting issue in the next review phase.
>
> An overview of our revision plan based on all reviewers’ feedback is as follows:
> * We will better clarify the assumptions, parameters, and implications of theorems in Section 4.
> * We will present a full discussion of limitations and future directions in a new Section 6.
> * We will add numerical results that better explains the method’s scalability and the interplay between $n$ and $d$.
> * We will add explanatory sentences as promised in our rebuttals to reviewers, and be sure to clear all typos noted.
>
> For papers mentioned below that are cited in [name, year] format, the citations are available in our manuscript’s reference; if instead a paper is cited by number such as [1], we will provide the complementary reference at the bottom. We now begin our responses to your questions.
>
> ---
>
> ***W1 & Q3. Algorithmic guarantees***
>
> We thank the reviewer for this thoughtful comment. The current manuscript aims to propose a new framework for nonparametric empirical Bayes and to analyze its statistical guarantees. We agree that a more thorough discussion of the algorithmic aspects of our approach is valuable and acknowledge that our current manuscript does not fully address this important point. We will explicitly discuss these issues in the limitation section.
>
> In particular, we agree that assuming solvability of the nonconvex optimization problem is a strong assumption. In the existing literature, global optimality guarantees for training neural networks are often established under overparameterization, for example via the neural tangent kernel regime [1]. In such settings, the objective behaves almost linearly and can satisfy favorable conditions such as the Polyak–Lojasiewicz inequality. Incorporating this line of analysis that linking optimization theory with approximation error of deep neural networks and statistical estimation bounds would provide a more comprehensive understanding of the proposed method. We will include these points in our revised manuscript.
>
> ***Q1. Interplay between n and d***
>
> We kindly refer you to our responses to @DsAZ (W2) where we bear in mind your question in answering there and it should cover most aspects concerning such interplay. In short, we plan to incorporate an additional experiment that explicitly play around these dependencies.
> We are happy to further discuss if you find it insufficient to answer your question here.
>
> ***Q2. Understanding the over-shrinkage bias***
>
> Your intuition from the information-theoretic perspective is certainly correct. The example demonstrates that the posterior mean (or one-step Tweedie) estimator $\mathbb{E}[\theta |X]$ is not sufficient for explaining the full variation in the distribution of $\theta$. Indeed, using the total law of variance, we have:
> $ Var(\theta) = Var[\mathbb{E}[\theta | X]] + \mathbb{E}[Var(\theta | X)]$. This shows that the posterior distribution of random variables $\mathbb{E}[\theta |X]$ can only partially explain the actual variation in $\theta$, and thus leads to over-shrunk estimators for $\theta_i$.
>
> In contrary, if we apply the multi-step Tweedie, then on one hand the ODE defines a deterministic flow path that “transport” the observation distribution to the (approximated) latent distribution, thus we can expect the resulting denoised estimator to distribute according to the latent distribution without any reduced variance. On the other hand, related to the above analysis, we may again understand from the information-theoretic perspective. Assume we know the true latent distribution and follow Equation (6) in our paper to denoise, then the denoising sequence of random variables $\\{X_i\\} = X^{(t=1)}, X^{(1-\Delta t)}, \dots, X^{(\Delta t)}, X^{(0)} = \\{\theta_i\\}$ defines a Markov chain connecting latent $\theta$ and observables $X_i$ and we wish to explain the variability in $\theta_i$ through the sequence. Then by the data processing inequality, we have $Var[\mathbb{E}[\theta | X^{(1)}]] \leq Var[\mathbb{E}[\theta | X^{(1-\Delta t)}]] \leq \cdots \leq Var[\mathbb{E}[\theta | X^{(\Delta t)}]]$, i..e, we gain enhanced explanatory power as we move along the ODE path. This clearly demonstrates why we prefer the multi-step procedure.
>
> Provided more space in the camera-ready version, we will attempt to incorporate the above analysis into our final manuscript.
>
> We would be happy to further discuss any questions or concerns that may arise during the discussion phase.
>
> ---
> [1] Jacot, Arthur, Franck Gabriel, and Clément Hongler. "Neural tangent kernel: Convergence and generalization in neural networks." Advances in neural information processing systems 31 (2018).

---

> > ### Comment · Reviewer_KpwC · 2025-08-05
> >
> > I'd like to thank the authors for their careful responses. They addressed the points I raised; in particular, I found the answers regarding the interplay between $d$ and $n$ satisfying. I trust the authors will include these considerations in the final version of the paper. That said, I remain somewhat unsure about the assumption concerning the solvability of the infinite-dimensional nonconvex problem. For now, I would like to keep my original score.

---

> ### Comment · Area_Chair_vMa5 · 2025-08-05
>
> Dear reviewer,
>
> Please read the authors' rebuttal if you haven't done so and state your response accordingly.
>
> Best,
> AC

---

> ### Author Response · Authors · 2025-08-07
>
> Dear reviewer KpwC,
>
> Thank you for your responses. We would like to further address your concerns over the solvability of the infinite-dimensional nonconvex problem.
> Firstly, when a parametrized class such as a deep neural network is used to approximate $G$, the problem reduces to optimizing over a finite-dimensional space. By the universal approximation theorem of neural networks [1], the global minimizer $\widehat{G}$ in Eq. (8) belongs to (or can be approximated arbitrarily well by) the parametrized class if a sufficiently expressive NN architecture is chosen.
>
> Regarding nonconvexity, the Polyak-Lojasiewicz (PL) condition is often satisfied by mean squared loss optimized over overparametrized neural networks, as shown in Liu et al. [2]. This condition guarantees existence of global minimizers and convergence of (stochastic) algorithms beyond convexity. Our objective Eq. (8), with normalizing flows parametrization of $G$, is a squared loss over an overparametrized NN, thus is expected to satisfy the PL condition and be solvable to global optimality.
>
> Recall that a function $f$ satisfies PL if for a positive constant $\mu>0$ the following is satisfied: $\\|\nabla f(x)\\|^2 \geq \mu \cdot (f(x) - f^\*)$. While theoretically proving PL of a particular problem is generally difficult due to sensitivity to NN architecture, we are able to present you with some empirical evidence that can help confirm the PL condition in our case. We fix a synthetic circle dataset with $d=8$ and $n=10^7$. A large number of samples help eliminate statistical instability.  In the table below we recorded along two training paths the minimum value of PL certificate achieved in so far iterations (PL LB), as well as approximated FD metric (as a proximal certificate of optimality). The PL certificate is defined via: $\frac{\\|\nabla_\theta \mathcal{J}_n(G(\theta))\\|^2}{\mathcal{J}_n(G(\theta))}$. Here $\theta$ represents the parameters in the parametrized model of $G$, $\mathcal{J}_n(G)$ is the objective function. Based on the above theory, if the certificate metric is constantly bounded away from a positive threshold $\mu > 0$ along the training trajectory, then we have good confidence that our objective should satisfy PL condition at least over the regions where the iterates traverse, and won’t get stuck in a local minimum.
>
> | Trial / Epoch | 2 |5  | 10 | 20   | 37 | 40 | 45 | 50 |
> |-|-|-|-|-|-|-|-|-|
> | **Trial 1** |  |    |    |    |  |    |  |   |
> | PL LB | 0.867 | 0.470 | 0.0869| 0.0328| 0.0117| 0.0117 | 0.0117 | 0.0117 | 0.0117 | 0.0117 |
> | FD | 0.0456| 0.0203| 0.0094  | 0.0102  | — | 0.0049 |  0.0046| 0.0045     |
> | **Trial 2**   |     |      |    |    |    |     |     |   |
> | PL LB  |0.821  | 0.178  | 0.111| 0.0440| 0.0157| 0.0157 | 0.0123 | 0.0123 |
> | FD | 0.3892  | 0.0601  | 0.0274  | 0.0109  | —     | 0.0084 |   0.0046 |  0.0051 |
>
> As shown, the PL certificate remains constantly lower bounded (0.0117 or 0.0123), well above $10^{-2}$, as optimization reaches near stationarity around epoch 40 ($1$ epoch $\\approx 10^4$ gradient steps). Using the Trial 1 epoch 50 checkpoint, we denoised a validation dataset. The denoised samples visually match the true circle closely and achieve MSE 0.1281 to clean samples, compared to oracle MSE 0.1270 (less than 1% optimality gap) and indicating sufficient global optimality.
>
> We also offer some theoretical intuition. Each squared score matching term in Eq. (8), under parametrization $G = G_\theta$, takes the form $\\| \nabla_x \log p_\theta(x) - y(x)\\|^2$. With some calculation, its PL certificate:
> $$
> \\left\\| \\frac{\\partial}{\\partial \\theta}\\| \\nabla\_x \\log p\_\\theta(x) - y(x)\\|^2 \\right\\|^2 / \\| \\nabla\_x \\log p\_\\theta(x) - y(x)\\|^2
> =4 \\| \\mathbb{E}\_{\\mu \\sim G\_\\theta} [\\varphi\_{\\Sigma}(x - \\mu) [\\nabla\_x \\log \\varphi\_{\\Sigma}(x - \\mu)]^\\top \\nabla\_\\theta \\log g\_\\theta(\\mu)]\\|^2
> $$
> Thus, the PL certificate is closely linked to the Fisher information $I\_{\\theta}(G) = \\mathbb{E}\_{G\_\\theta}\\|\\nabla\_{\\theta} \\log g\_{\\theta}(\\mu)\\|^2$.
> The full objective satisfies PL if a weighted average of such terms is bounded below, which will hold necessarily upon the FI remains positive. This implies a meaningful and trainable neural parametrization and aligns with the empirical observation of nonvanishing gradient.
>
> While the above arguments are not fully rigorous due to time and space constraints, we hope they provide confidence in the solvability of our objective. We also emphasize that our work focuses other important challenges in empirical Bayes.
>
> [1] Hornik, Kurt, Maxwell Stinchcombe, and Halbert White. "Multilayer feedforward networks are universal approximators." Neural networks 2.5 (1989): 359-366.
>
> [2] Liu, Chaoyue, Libin Zhu, and Mikhail Belkin. "Loss landscapes and optimization in over-parameterized non-linear systems and neural networks." Applied and Computational Harmonic Analysis 59 (2022): 85-116.

---

> > ### Comment · Reviewer_KpwC · 2025-08-07
> >
> > I would like to thank the authors for their detailed response. The empirical results do offer some indication and intuition as to why the issue might be resolved. That said, I recognize that your work addresses other important challenges. I will think about whether to update my score to reflect that.

---

> > > ### Author Response · Authors · 2025-08-08
> > >
> > > Dear reviewer KpwC,
> > >
> > > Thank you for your careful review and for acknowledging the broader contributions of our work. We’re grateful that you are considering revisiting your score and appreciate the constructive feedback throughout the process.

---

### Official Review · Reviewer_DsAZ · 2025-06-29

**Clarity:** 3
**Significance:** 3
**Originality:** 4
**Rating:** 4
**Confidence:** 3

**Summary:**

The paper propose Score-based Diffusion with G-Modeling for non-parametric empirical Bayes in multivariate heteroscedastic Gaussian‐location mixtures. The key idea is to reinterpret each noisy observation as the time-1 snapshot of a forward SDE whose latent initial state is drawn from an unknown prior G. This perspective yields a reverse-time ODE (Eq. 7/9) that supplies a multi-step denoising rule generalising Tweedie’s one-shot formula. For the homoscedastic case the authors prove that the plug-in estimator achieves near-parametric Fisher-risk and Hellinger error O(polylog(n)/n) and Wasserstein error O(1/logn) (Thm 1). They extend the guarantees to heteroscedastic noise (Thm 2). Two practical solvers are described: a discrete-support EM-style optimiser and a continuous normalising-flow parameterisation. Experiments on synthetic and real-world data show improvement over NPMLE baselines.

**Questions:**

See weaknesses above for points about the paper which I am concerned about. I would appreciate clarifications on them to increase my score.

**Ethical Concerns:**

["NO or VERY MINOR ethics concerns only"]

**Final Justification:**

I trust the authors to include the explanations about the dependence on the ambient dimension and their rationale for not comparing with other diffusion-based mixture learners as baselines since their problem setting is different. I would therefore like to maintain my original score.

**Limitations:**

Yes

**Quality:**

3

**Strengths And Weaknesses:**

Strengths
1. They paper proposes a reverse-time denoising procedure and score-matching objective that yield near-parametric Fisher-risk and Hellinger error rates, improving on prior diffusion-model analyses as well as NPMLE estimators.

2. Practical denoising advantage – Multi-step reverse denoiser corrects the classic Tweedie over-shrinkage and demonstrates tangible MSE gains in experiments.

3. Empirical results – On synthetic and real data, SDGM achieves lower Fisher divergence and negative log-likelihood than NPMLE and PEM baselines, supplementing their theory results.

Weaknesses
1. Requires compact support, well conditioned heteroscedastic covariances, but it is not clear why the assumption holds in practice.

2. Error bounds carry $log^d(n)$ factors; the actual exponent is unspecified, so guarantees may become impractical for large $d$.

3.Each gradient step touches all n observations (or all obsevations in the batch); the paper provides no wall-clock or memory benchmarks, leaving practical scalability unclear. In context of Diffusion models, there was a recent [paper ](https://arxiv.org/pdf/2310.12395) which tried to come up with a non-parametric form of the score function, but they faced certain challenges when compared with neural networks regarding scalability and overfitting. I am curious to see if the rich representation power of neural networks, that has been applied very successfully with diffusion models for score representation, could be applied here as well.

4. Experiments do not compare against recent diffusion-based mixture learners mentioned in the manuscript. It would help to state why these methods don't apply to the authors' experiments directly.

---

> ### Author Rebuttal · Authors · 2025-07-30
>
> Dear reviewer **DsAZ**,
>
> We sincerely thank you for recognizing the technical strengths of our proposal. Indeed, the novel integration of the diffusion reinterpretation with the empirical Bayes G-modeling naturally permits both efficient score-based learning of latent measure and the multi-step Tweedie-type denoising procedure as coherent ingredients. A classical score-based approach with direct parametrization of the score function cannot immediately resolve the two challenges in empirical Bayes we focused on. Our results demonstrated that the proposed framework tackled to a satisfactory extent the over-shrinkage bias and scalability challenges of traditional likelihood-based methods.
>
> We also want to thank reviewers for bringing constructive feedback based upon which we would like to further improve our manuscript. For each question, we intend to both clarify our understanding, and discuss plans to update our manuscript if we have the opportunity to submit a final version. Due to space limits, we may need to refer you to our responses to other reviewers’ related questions. We apologize for such inconvenience and we are happy to follow up with any persisting issue in the next review phase.
>
> An overview of our revision plan based on all reviewers’ feedback is as follows:
> * We will better clarify the assumptions, parameters, and implications of theorems in Section 4.
> * We will present a full discussion of limitations and future directions in a new Section 6.
> * We will add numerical results that better explains the method’s scalability and the interplay between $n$ and $d$.
> * We will add explanatory sentences as promised in our rebuttals to reviewers, and be sure to clear all typos noted.
>
> For papers mentioned below that are cited in [name, year] format, the citations are available in our manuscript’s reference; if instead a paper is cited by number such as [1], we will provide the complementary reference at the bottom. We respond to your questions and concerns as follows.
>
> ---
>
> ***W1. Whether assumptions are realistic***
>
> Due to space constraints, we kindly refer you to our responses to @9mdT (W1). If you find it unsatisfactory for your question, we are happy to follow up in the next phase.
>
> ***W2. Dependency on the ambient dimension $d$***
>
> We presented $O(d)$ in the exponent of $\log n$ with the aim to reduce burden to understand the core takeaways of the theorem. Our technical supplemental provides the exact exponent as $(5d+2)/2$. This matches previous work with a similar exponent order (Corollary 3.2 of [Saha and Guntuboyina, 2020] gives 2d or 3d under various comparable assumptions).
>
> We agree with the reviewer’s concerns that the bound $(\log n)^{O(d)}/n$ becomes impractical for very large $d$. This on one hand reveals the fundamental curse of dimensionality difficulty of the task: We estimate a $d$-dimensional measure and denoise each $\theta_i$ from a single observation $X_i$. In fact, Kim and Guntuboyina [1] show the minimax rate in Hellinger distance for estimating Gaussian location mixtures is bounded from below by $(\log n)^d/n$, confirming the difficulty of the task.
>
> On the other hand, our theorems attempted to demonstrate a fair comparison under comparable assumptions as these work. Such rate is not available in literature before our work for score-based empirical Bayes approach in high dimensional setting.
> As we mentioned in Related Work (Line 131), whether a non-NPMLE method can achieve a comparable rate with leading $O(1/n)$  factor is only known recently in Ghosh et al. [2025], and we are the first work that extends their $d=1$ setting to arbitrary dimension $d\geq 1$. In addition, establishing our results is nontrivial, requiring carefully adapting, reinventing, and integrating results from various related fields. Consequently, we believe that our results are significant, which presents a baseline guarantee and, together with our numerical findings, further arouses interest if we could attain better rates if further structural information is leveraged.
>
> Indeed, our experiment results (Figure 2) suggest that a score-based approach is more robust in computational scalability than solution methods of NPMLE in higher dimensions. Based on our experiments and recent belief and observations in literature that diffusion models can identify data manifolds (e.g., in [2]), an educated conjecture is that, if we indeed know the intrinsic dimension $d_0$ of the low-dimensional manifold, instead of the ambient dimension $d$, then we may be able to derive sharper results that avoid the dependence on $d$ in the exponent of the logarithmic factor.
> Following reviewers’ suggestions, we conducted additional experiments to investigate two questions: (1) if our method exhibits consistent scalability in higher dimension, and (2) if the empirical interplay between $n$ and $d$, with a particular focus on the latter, performs at least better than the theoretical convergence rate.  We constructed the same synthetic circle dataset appeared in our Sec. 5 with a fixed $n=10^7$ and varied $d = 8, 16, 32$. We applied our SDGM-C, running 5 trials of 6 full data passes for each setting. We report two metrics from our paper: the Fisher divergence (FD) and the negative log-likelihood (NLL), with their normalized values to facilitate comparison across $d$.
>
> | metric \ d | 8 | 16 | 32 |
> |----------|----------|----------|----------|
> |  NLL   |   13.525   |  24.824    | 47.456  |
> | (normalized)   |  1 |    1.84   | 3.51|
> | |
> | FD   |   0.018   |   0.024    |  0.036 |
> | (normalized)    | 1  |  1.36    |  2.05 |
>
> The results show both metrics scales (mostly) linear with $d$ in contrast to the unfavorable $(\log n)^d$ suggested by the baseline theory.  It may motivate the hypothesis of an improved rate such as $C d{(\log n)^{d_0}}/{n}$.
>
> For theoretical justification on improving risk rates, we look forward to better leveraging the structure and exploring various different proof tools as a future work.
> For the current manuscript, based on the above discussions, we plan to (1) revise the theorem implication paragraph (Lines 222 to 242) and add the higher dimensional experiments with varying $(n, d)$ in Section 5 to better demonstrate and interpret the scalability issue, and (2) fully explain the theoretical limitations and envision the future directions in the added section.
>
> ***W3. Resource benchmark and neural network***
>
> We appreciate that the reviewer shares the insightful paper with us. It looks like their goal is to circumvent training, while our method, though nonparametric, did perform parametrization of $G$ and requires training. In essence, the term “nonparametric” as in various literature means that we do not assume any parametric form of the true target (i.e., $G^\*$ is not assumed to belong to any specific parametrized measure class), but we can use various parametrized models to serve the learning task.
> Indeed, our implementation already makes use of deep neural networks and benefits from GPU speedup. As we explained in Section 4, the normalizing flows are a class of neural networks that we used to parametrize $G^*$ continuously and our SDGM-C tends to perform better than the discrete counterpart.
>
> For a quick glimpse into the resource benchmarks, we report  average wall-clock time per step and the peak GPU memory usage over the training tasks with batch size 1024 and 500k Pytorch model parameters performed on 2x Nvidia V100 GPUs:
> | d  | Time (second per step) | Peak GPU usage (GB) |
> |----------|----------|----------|
> | 16    |    0.53      |    2.0     |
> | 32    |     1.17     |     5.5     |
>
> For a very rough comparison, the statistics reported in the official repo for [Song et al., 2021a] shows their Pytorch model runs 0.56 second per step and uses 20.6 GB memory in total on 4x V100 GPUs.
> We believe there exists room for optimizing our implementations and we expect to explore these practical optimization considerations as a future work.
>
> ***W4. Lacking other diffusion-based mixture learners as baselines***
>
> We did not compare with diffusion-based mixture learners [Shah et al. 2023, Kim and Ye 2022] primarily because of two reasons.
> * Resultant denoisers from these methods can only be attained through the one-step Tweedie and the unfavorable over-shrinkage bias is unavoidable.
>  These methods are not compatible with the multi-step denoising procedure, due to they all directly parametrize the score function without G-modeling. As we mentioned in Line 180, without G-modeling,  scores are not defined between $t = 0$ to 1 and are not learnable because no data exists in the time interval. It is implausible for the score network learned in forward time ($t > 1$) to interpolate for the backward time ($t < 1$), thus the multi-step denoising steps would not work.
> * All these learners do not learn the latent distribution. By learning the scores directly, they only permit efficient generation of samples from the noisy mixture, yet no $\widehat{G}$ can be recovered. In such cases,  half of our metrics (NLL and W2) are uncomputable and cannot be reported for a fair comparison.
>
> Given the above reasons, we believed that while relevant to our work, they are too brute-force for serving the core task, which has evenly weighted focuses on both the recovery of latent measure and on denoising the observations. These approaches didn’t excel in either task. Consequently, we didn’t include them as a valid baseline. If the reviewer finds some strong reasons to include them, we are happy to further discuss them.
>
> ---
> [1] Arlene K. H. Kim, Adityanand Guntuboyina "Minimax bounds for estimating multivariate Gaussian location mixtures," Electronic Journal of Statistics, Electron. J. Statist. 16(1), 1461-1484, (2022)
> [2] Pidstrigach, Jakiw. "Score-based generative models detect manifolds." Advances in Neural Information Processing Systems 35 (2022): 35852-35865.

---

> > ### Comment · Reviewer_DsAZ · 2025-08-04
> >
> > Thank you for your response. I trust the authors to include the explanations about the dependence on the ambient dimension and their rationale for not comparing with other diffusion-based mixture learners as baselines. I would like to retain my original score.

---

> > > ### Author Response · Authors · 2025-08-05
> > >
> > > Thank you for your response. We sincerely appreciate your valuable suggestions and will carefully incorporate your feedback into our manuscript. Many thanks once again!

---

### Official Review · Reviewer_fmp1 · 2025-06-30

**Clarity:** 3
**Significance:** 3
**Originality:** 3
**Rating:** 4
**Confidence:** 3

**Summary:**

This paper proposed a non-parametric empirical Bayes approach, which is based on a generalized score-based diffusion framework, for learning multivariate Gaussian mixture models with homoscedastic or heteroscedastic noise. The main idea is to view each observation as temporal slices of a family of stochastic diffusion process, which enables a multi-level score-based objective for learning the latent location distribution. Theoretically, a near-parametric error rate was established for the score estimation error. They also show that the proposed method outperforms the non-parametric maximum likelihood estimator in both density estimation and denoising fidelity.

**Questions:**

-	In the real-world astronomy experiment, what constitutes an astrochemically meaningful structure in the denoised output? Is the goal to recover scientifically interpretable features, or is sharpness alone the main criterion for success? Some clarification on the downstream relevance would help assess the practical value of the method.
-	The constants hidden in Theorems 1 and 2 are stated to depend on the dimension ddd. Could you specify how this dependence scales (e.g., linearly, polynomially, or exponentially)? Also, how does the method behave in higher dimensions? The experiments are limited to $d=19$ -- do you anticipate scalability to much higher dimensions?
-	How is the upper bound $T$ for the diffusion time chosen in practice? Does its value affect the convergence rate or the bias-variance tradeoff in the score estimation? Any practical heuristics or theoretical guidance would be appreciated.

**Ethical Concerns:**

["NO or VERY MINOR ethics concerns only"]

**Final Justification:**

The author has adequately address my concerns, and I maintain my positive evaluation of this paper

**Limitations:**

The main limitations are the computation of the marginal density $f$. Additionally, the paper lacks discussion and empirical validation of performance in higher-dimensional settings, limiting insight into the method’s scalability.

**Quality:**

3

**Strengths And Weaknesses:**

Strength:

- The use of score matching, rather than the classical NPMLE, to estimate the prior within the empirical Bayes framework via G-modeling is novel and insightful.
- The introduction of a multi-step denoising procedure based on reverse-time diffusion dynamics offers a principled generalization of the traditional one-step Tweedie estimator, mitigating over-shrinkage bias.
- The theoretical analysis is rigorous and well-developed, providing near-parametric guarantees for score estimation and extending to the more realistic heteroscedastic noise setting.

Weakness:

- While the overall methodological development is well-presented, the paper lacks a clear explanation of how the marginal density $f_{G,\Sigma_i}$  computed in practice. Given its central role in both the score matching objective and the denoising procedure, a more explicit description would be helpful for reproducibility and understanding.
- The rationale for preferring score matching over nonparametric maximum likelihood estimation (NPMLE) is not fully justified. As mentioned around Line 233, the convergence rate achieved by score matching merely matches that of NPMLE. Moreover, it seems plausible that NPMLE could also be applied to perturbed observations across multiple noise levels, as in the multi-step denoising setting. Since score matching still relies on computing the marginal score, it is not immediately clear what practical advantages it offers in this context. A deeper comparison—both theoretical and empirical—would strengthen the case for using score-based methods.

Typo:
- Line 69: scoring matching  -> score matching

---

> ### Author Rebuttal · Authors · 2025-07-28
>
> Dear reviewer **fmp1**,
>
> We sincerely thank you for your kind words about our novelty and technical strengths. Our idea of integrating diffusion reinterpretation with empirical Bayes G-modeling jointly and naturally permits both efficient score-based latent learning and multi-step Tweedie-type denoising. This cannot be achieved with a classical score-based approach with direct parametrization of the score function. Our results demonstrated that the proposed framework did tackle the over-shrinkage bias and scalability challenges of traditional likelihood-based methods.
>
> We also want to thank you for bringing constructive suggestions based upon which we would like to further improve our manuscript. For each question, we intend to both clarify our understanding, and discuss plans to update our manuscript if we have the opportunity to submit a final version. Due to space limits, we may need to refer you to our responses to other reviewers’ related questions. We apologize for such inconvenience and we are happy to follow up with any persisting issue in the next review phase.
>
> An overview of our revision plan based on all reviewers’ feedback is as follows:
> * We will better clarify the assumptions, parameters, and implications of theorems in Section 4.
> * We will present a full discussion of limitations and future directions in a new Section 6.
> * We will add numerical results that better explains the method’s scalability and the interplay between $n$ and $d$.
> * We will add explanatory sentences as promised in our rebuttals to reviewers, and be sure to clear all typos noted.
>
> For papers mentioned below that are cited in [name, year] format, the citations are available in our manuscript’s reference. We respond to your questions and concerns as follows.
>
> ---
>
> ***W1. Computation of marginal density***
>
> The computation of the marginal density and its log-gradient is straightforward in practice. If $G$ is represented by a discrete measure with finite number of supports $\sum_{j=1}^m w_j \mu_j$ (strategy one in Section 4), then $f_{G, \Sigma_i}(x) = \sum_{j=1}^N w_j \varphi_{\Sigma_i}(x - \mu_j)$ where $\phi_{\Sigma}(\cdot)$ is the standard zero-mean multivariate Gaussian density function with covariance $\Sigma$ that we gave in Line 38, and $f_{G,\Sigma_i}$ is basically sum of exponentially transformed matrix products. Similarly we can obtain $\nabla_x f_{G, \Sigma_i}(x) =  -\sum_{j=1}^N w_j \varphi_{\Sigma_i}(x - \mu_j) \Sigma_i^{-1} (x - \mu_j)$ and the score $\nabla \log f_{G,\Sigma_i} = \nabla_x f_{G, \Sigma_i} /  f_{G, \Sigma_i}$ via chain rule, which is a weighted average of vectors $\Sigma_i^{-1}\{x - \mu_j\}$.
> As we explained in Section 4, if $\widehat{G}$ is represented by a continuous normalizing flow, Monte Carlo sampling can serve as a theoretically backed tool to approximate the density, and the computation can be done with the same formulas above.
> Note that the computations involve mostly matrix operations and can be efficiently performed on GPU with proper tensorization. Our code in supplementary presents how these steps are actually performed and allows reproducibility.
> We plan to explicitly display the above formulas in Sec. 4 to assist understanding.
>
> ***W2. score matching vs. NPMLE***
>
> Overall, we believe that either score matching or NPMLE contributes as a component of a full empirical Bayes framework. We believe that it is our overall framework from a stochastic diffusion perspective that stands out, with the score matching as a coherent and natural ingredient. We are also deeply interested in the possibility to improve solution methods of NPMLE by leveraging established knowledge from literature and our current manuscript.
>
> For our defense on theoretical convergence rate, please refer to our responses to @DsAZ (W2) where we jointly answered some concerns over how we interpret the rate as a function of $d$. In short, our theoretical analysis aims at achieving the rate of convergence under comparable assumptions as existing work. The same order of dependence on $d$ reveals the fundamental difficulty of the task. Empirically, we observe improved scalability compared to SOTA solution methods for NPMLE (as visible in Figure 2). We believe in possibilities to derive sharper theoretical results if we leverage more structural information.
>
> Regarding the multi-level perturbation for NPMLE, while it seems plausible, it is not clear how NPMLE can benefit from such perturbation. It does not seem to solve either important issue faced by likelihood-based approaches we detailed in Introduction.
> On the other hand, perturbation is very natural for score learning, and is justified theoretically and proven useful in diffusion literature [Vincent 2011, Song et al. 2021a]. Intuitively, score matching aims at learning a vector field over space thus perturbation is beneficial for exploration spanning the whole space.
>
> Further, whether the multi-step denoising is successful heavily relies on accurate estimation of the latent measure. This is because the reverse-time ODE effectively defines a “transport” mapping from observation to the latent measure. However, as our numerical results demonstrate (Fig. 2), existing NPMLE solvers are highly unscalable and the estimated $\widehat{G}$ is very low-quality even in $d=4$. On the other hand, our numerical results suggest that score-based approach indeed achieves practical advantages in both numeric metrics and visible denoising effects.
>
> We plan to dedicate a paragraph in the Limitations and Future Work section that summarizes the above discussion and compares the two approaches in detail. We hope our work not only presented a new method for empirical Bayes, but also shed some light on the advances in likelihood-based approaches.
>
>
> ***Q1. Interpretation of astronomy experiment***
>
> One particularly relevant downstream application appears in the cited work [Ratcliffe et al., 2020]. The goal there is to cluster the stars by their chemical similarity using the same 19-dimensional APOGEE chemical abundance dataset, and then examine the resulting groups in age and galactic position to trace disk assembly over time. Before clustering, they quantify pairwise correlations and apply PCA/Isomap to identify influential features and prevent overfitting in high dimensions, yet one difficulty they mentioned explicitly (Sec. 4.2) in the paper is that large uncertainties (i.e., noise we aim to reduce) in the measurements reveal extremely weak correlations among the elements. On the other hand, it is assumed that “our data lie on an embedded nonlinear manifold within the 19-dimensional space” (Sec. 3.1.3).
>
> These points suggest that a denoising step that reduces measurement uncertainties and recovers the manifold geometry can improve the estimation of true abundance correlations and help downstream clustering mitigate over‑fitting due to noise and strong feature correlations. Consequently, we believe that while sharpness may not necessarily be a criterion for success, the visibly thin, coherent manifolds recovered from the denoised data in our Figure 4 suggests a success, following the above astronomers’ assumptions. Such qualitative criterion is used widely in cited NPMLE works that worked on this data [Soloff et al., 2025, Zhang et al., 2024]. Notice that a clear manifold structure can be successfully recovered in prior work only if they analyze a 2-dimensional projection (e.g., Figure 6 of [Zhang et al., 2024]), while our work successfully applies over the original 19 dimensional space.
>
> For our final version, we plan to add the following sentence after Line 333 to better demonstrate the practical value of our method:
> “In particular, successful denoising and recovery of low‑dimensional manifold structure in chemical abundance space are crucial for downstream tasks such as clustering stars by mitigating over‑fitting driven by measurement noise and strong inter‑element correlations [Ratcliffe et al., 2020].”
>
>
> ***Q2. Scalability in high dimensional settings***
>
> We once again refer you to our responses to @DsAZ (W2) and @9mdT (W1) for discussion about theoretical scalability. For computational scalability, our method is proved to be much better compared to SOTA NPMLE solvers. Recent paper [Zhang et al. 2024; Fig. 14] (our baselines) reported their struggles even in computing a nontrivial solution for $d\geq 10$. Our results (e.g. Fig. 2) confirms visible recovery and denoising in $d=16$. The additional experiment in our response to @DsAZ (W2) scales to $d=32$. Thus, we believe in a positive answer to your question.
>
> ***Q3. Choice of upper bound $T$***
>
> Recall that $T$ controls how further we wish to smooth the empirical density in the training objective. We did require an upper bound on $T$ to derive the desired guarantees in Theorem 1 & 2, which was shown in the Appendix. Your intuition is right: the core role $T$ plays is indeed related to the convergence rate of the smoothed empirical density from its population counterpart.  It in turn affects the score estimation error. Technically, we pick an upper bound so as to ensure balance among error terms. Intuitively, it follows the same philosophy as in kernel density estimation where stronger smoothing ($T$ larger) leads to both empirical and population distributions collapsed to flatter densities thus the variance reduces, yet we lose more information over the high-density regions leading to high bias. In our final version, we will specify the upper bound more explicitly in the main text.
>
> In our experiments, guided by the above theoretical intuition, we heuristically found that $T$ need not be set super large for the training to work successfully so we fix $T$ at 1.5 for our experiments. Such hand design is ubiquitous in diffusion literature, e.g., [Song et al., 2021a].
>
> We would love to further discuss any follow-up questions during the incoming phase.

---

> > ### Comment · Reviewer_fmp1 · 2025-08-07
> >
> > Thanks for your response! Just a minor issue regarding the computation of the marginal density when $G$ is a continuous distribution. Monte carlo estimates can be use, but when taking the log, the score would be biased. How many samples are usually required to provide good approximation then?

---

> > > ### Author Response · Authors · 2025-08-08
> > >
> > > Dear reviewer fmp1,
> > >
> > > Thank you so much for your responses! We are happy to follow up with the issue over discrete approximation of scores.
> > >
> > > We first give our empirical answer to your question on *How many samples are usually required to provide a good approximation?* Across all our experiments with dimension $d \\in [8, 32]$, we found out that the choice of 2,000 samples is enough for consistent positive results. For example, as we showed in our responses to Reviewer @KpwC, such a choice achieved a denoising performance with less than 1% optimality gap compared to oracle denoiser (knowing the true $G$) for the circle problem in $d=8$. The choice of sample size over $G$-model is of the same order of magnitude as the data batch size (1024) and we didn’t tune the hyperparameters extensively.
> > > Because all our experiments are constructed or believed to have low-dimensional compact manifold structure, it is conceivable that no massive sample is required. For example, for a measure $G$ distributed over a circle (Example 1 in Sec. 5) embedded in higher-dimensional space, we only need a discrete measure to capture the variations in the first two intrinsic coordinates, and pad zero over the remaining ones.
> > >
> > > We now discuss the problem of *bias* and scaling with an increasing number of samples to ensure good approximations. As the reviewer has observed, the Monte Carlo sampling yields a self-normalizing ratio estimator of the score function.
> > > The systematic characterization of bias of ratio estimators is well studied in statistical literature. In particular, following a standard first-order Taylor expansion, the bias of the Monte Carlo score estimator compared to the population score decays at a linear rate $O(1/m)$ in expectation as we increase the number of samples $m$. Regarding model training with the biased estimator, recent papers established convergence results with biased gradient [1], and it is known that optimization bias can have beneficial implicit regularization effects to help generalization [2, 3]. Restricting a properly small sample size may also promote the model towards a desired low-complexity geometry.  For inference (i.e., denoising), both the magnitude and angle of the score vector is biased of the order $O(1/m)$. In practice, we observe that the denoised samples form a shape close to the learned latent model, so we believe that the bias is not a significant source of error.
> > >
> > > In summary, our experiments demonstrate the sampling bias appears mild with a moderate number of samples and we present some theoretical intuition to justify the empirical evidence. We don’t claim optimal design of our empirical approach and see improvements as future work. We will discuss these numerical issues in Limitations. We once again thank you for your careful review of our work in detail.
> > >
> > > [1] Ajalloeian, Ahmad, and Sebastian U. Stich. "On the convergence of SGD with biased gradients." arXiv preprint arXiv:2008.00051 (2020).
> > > [2] Wu, Lei, and Weijie J. Su. "The implicit regularization of dynamical stability in stochastic gradient descent." International Conference on Machine Learning. PMLR, 2023.
> > > [3] Smith, Samuel L., et al. "On the origin of implicit regularization in stochastic gradient descent." arXiv preprint arXiv:2101.12176 (2021).

---

> ### Comment · Area_Chair_vMa5 · 2025-08-05
>
> Dear reviewer,
>
> Please read the authors' rebuttal if you haven't done so and state your response accordingly.
>
> Best,
> AC

---

> ### Author Response · Authors · 2025-08-07
>
> Dear Reviewer fmp1,
>
> I hope this message finds you well. We truly appreciate the time and effort you’ve dedicated to reviewing our work, and we hope our rebuttal helped clarify your concerns.
>
> We want to gently follow up to see if you might be able to share any further comments before the discussion phase concludes.
>
> Thank you again for your thoughtful feedback and for supporting the review process of our paper. We look forward to hearing from you.
>
> Best regards, authors of submission 26547

---

### Official Review · Reviewer_9mdT · 2025-07-03

**Clarity:** 3
**Significance:** 3
**Originality:** 4
**Rating:** 5
**Confidence:** 2

**Summary:**

This paper presents a novel framework that integrates score-based diffusion models with G-modeling for empirical Bayes estimation, particularly targeting heteroscedastic Gaussian mixture models. The authors reinterpret each noisy observation as a terminal state of a diffusion process, enabling the estimation of a nonparametric prior distribution $G^*$ without assuming parametric forms.

**Questions:**

I have some questions regarding Theorem 1:

1. The requirement to target distribution is only 'compact support'. If we have better property, like $G^*$ has strongly-convex score, how will the bound in Theorem improve?

2. The noise is supposed to be lower bounded by $\underline{\sigma}$. Do you require $\underline{\sigma}$ to be strictly positive? Or how the result depends on the condition number of $\Sigma_i$? Also, can you verify these assumptions by some examples in section 5?

**Ethical Concerns:**

["NO or VERY MINOR ethics concerns only"]

**Final Justification:**

All of my concerns have been addressed in the rebuttal and so I raised my score.

**Limitations:**

Yes.

**Quality:**

3

**Strengths And Weaknesses:**

The paper delivers a technically sound and mathematically sophisticated contribution to both diffusion modeling and empirical Bayes literature.The use of score matching in G-modeling is novel and theoretically grounded, with convergence results rigorously derived. As far as I know, this is the first work to extend score-based methods to a heteroscedastic Gaussian mixture setting with provable statistical guarantees.

The only weakness I see is that the main theoretical result (Theorem 1 in Section 3) is not fully interpretable in its current form: the constant dependencies (on $\sigma$) in the bound are not made explicit, making it difficult to assess practical implications or generality. Also, the assumptions are not verified or illustrated by the experiments. See details below in the Question part.

I would raise my score if the authors are able to (1) clarify the structure and implications of Theorem 1 with precise dependencies and (2) confirm whether the assumptions are satisfied in the presented synthetic and real-data examples.

---

> ### Author Rebuttal · Authors · 2025-07-30
>
> Dear Reviewer **9mdT**,
>
> We are grateful for your careful recognition of our framework. Indeed, the novel integration of the diffusion reinterpretation with the empirical Bayes G-modeling naturally permits both efficient score-based learning of latent measure and the multi-step Tweedie-type denoising procedure as coherent ingredients. A classical score-based approach with direct parametrization of the score function cannot immediately achieve the two challenges in empirical Bayes we focused on. Our results demonstrated that the proposed framework tackled to a satisfactory extent the over-shrinkage bias and scalability challenges of traditional likelihood-based methods.
>
> We also want to thank reviewers for bringing constructive feedback based upon which we would like to further improve our manuscript. For each question, we intend to both clarify our understanding, and discuss plans to update our manuscript if we have the opportunity to submit a final version. Due to space limits, we may need to refer you to our responses to other reviewers’ related questions. We apologize for such inconvenience and we are happy to follow up with any persisting issue in the next review phase.
>
> An overview of our revision plan based on all reviewers’ feedback is as follows:
> * We will better clarify the assumptions, parameters, and implications of theorems in Section 4.
> * We will present a full discussion of limitations and future directions in a new Section 6.
> * We will add numerical results that better explains the method’s scalability and the interplay between $n$ and $d$.
> * We will add explanatory sentences as promised in our rebuttals to reviewers, and be sure to clear all typos noted.
>
> For papers mentioned below that are cited in [name, year] format, the citations are available in our manuscript’s reference; if instead a paper is cited by number such as [1], we will provide the complementary reference at the bottom. We respond to your questions and concerns as follows.
>
> ---
>
>
> ***W1. Are assumptions realistic***
>
> The compact support assumption both covers a vast variety of realistic scenarios and is commonly adopted in prior works. Mixture model (1) is used when researchers believe there appears to be a certain structure to unveil from noisy data. In particular, true signals of our synthetic datasets are constructed to reside in low-dimensional compact manifolds (e.g., a 2D circle or letter “A”) embedded in a high-dimensional space.  The astronomical data is preprocessed by astronomers and statisticians in [Ratcliffe et al., 2020; Sec. 2.2]. Specifically, they retain only those stars that have abundance features confined within the box $[-1, 1]^{19}$, and treat the stars violating the rule as outliers due to anomalous measurements. This crucial step implies the scientific belief that the features of a "proper" star concerned in the downstream study should live in a compactly supported space within the box. We discuss more about the downstream relevance in our responses to @fmp1 (Q1).
> In short, we can certify that all examples in Section 5 indeed satisfy the compact support assumption. These examples are representative of the common scenarios that the model (1) applies to and researchers are interested in. We will make these statements explicit in our final version.
> While a mild generalization may be obtained for measures with decaying tails (see e.g., [Soloff et al., 2025]), the rates are vastly similar and we believe compactness is sufficient for conveying the core message of our work and is also satisfied for experiments we performed.
>
> Conditions on noise covariances are ubiquitous, important, and satisfied in our experiments. In various scenarios concerned by the model (1), including scientific measurements, covariance information is available or can be estimated from device-specific calibrated noise models, as we justified this in Line 21. For numerical examples in Section 5, the APOGEE dataset comes with known standard deviation estimates bounded within [0.01, 0.62] and has mean 0.05. Our synthetic examples are also constructed with random bounded variances, as shown in Line 316.  Thus, our examples all satisfy the assumption on the noise covariances.
>
> We now discuss why it is important to assume well-conditioned covariances. Intuitively, we require an upper bound because if $\overline{\sigma} \to \infty$, then it is information-theoretically impossible to extract useful knowledge from the data. On the other hand, a lower bound on $\underline{\sigma} > 0$ is needed to ensure the scores are well-defined everywhere across the space. Since we assume no parametric form of $G^\*$, it can potentially take a discrete form. Without any perturbation it is nondifferentiable at its component locations. To guard against these malicious possibilities, a positive yet finite noise scale is expected to smooth out degenerate locations and ensure well-behaved scores along an arbitrary SDE path for any $t > 0$.
>
> Furthermore, both the compact support and noise assumptions are commonly used in empirical Bayes literature [Saha and Guntuboyina 2020, Soloff et al. 2025] that proved convergence rates under these conditions. We did not invent any new conditions that are special or unrealistic. We presented these assumptions to enable comparison with existing established results.
>
> Regarding explicitness of constants, we would like to first overview the implicit part that involves $M, d, (\overline{\sigma}, \underline{\sigma}), g$. As we can imagine, the diffusion coefficient $g$ serves to rescale the covariance and thus can be absorbed in the $\sigma$ terms. Then we have roughly $C_{M, d, (\overline{\sigma}, \underline{\sigma})} \approx C\cdot d(M^2\overline{\sigma}/\underline{\sigma})^{d/2}$, which involves scaling due to condition number of covariances $(\overline{\sigma}/\underline{\sigma})$, and $M^d$ corresponds to covering the high-density region. Our supplemental contains relevant derivations in detail.
> We also want to emphasize that the near-parametric sample complexity part $\text{polylog}(n)/n$ is where most literature compares against and focuses on improving, while the leading constant dependencies are usually problem-specific and appear regardless of which learning method is adopted. Our risk presentation style is consistent with both empirical Bayes literature such as [Soloff et al. 2025] and various papers in this venue to highlight and convey the core takeaways. Sample complexity provides the core practical guidance on efficient utilization of collected data. We further discussed the sample complexity in our responses to @DsAZ (W2).
>
>
>
> ***Q1. Stronger latent assumptions***
>
> Indeed, if there exists better properties to leverage, it is possible to derive improved bounds. We interpret that your mentioned idea of “$G^\*$ has strongly convex score”  means that its density $G^\*(x) \propto \exp(-g(x))$ for some strongly convex function $g$. Notice that the score is a vector and it is ambiguous to define a “strongly convex” score. The score-based approach already leverages such convexity properties of Gaussian smoothing that allows efficient sampling and denoising due to good isoperimetric constants (see e.g., [1]). Unfortunately, we cannot think of any obvious way to leverage this additional structural information if the true latent measure is also strongly log concave.
>
> We do present some conjecture if we can improve the rate with the knowledge of intrinsic low-dimensional manifold dimension in our responses to @DsAZ (W2), which is supported by our experiments and the common belief that diffusion models can detect manifolds.
> Once again, we want to make assumptions consistent with existing empirical Bayes literature to deliver comparable results. We look forward to developing more advanced theories in future work. We will be sure to envision such directions in the new Limitations section.
>
> ***Q2. Covariance assumptions***
>
> We mentioned above that yes, we do require $\underline{\sigma}$ strictly positive for well-defined scores, and the dependence on the condition number. We have also certified the assumptions for our examples above.
> We will be sure to clarify $0 \prec \underline{\sigma} I_d \preceq \Sigma_i$ in our Assumption 1, and discuss the dependency of these problem specifications better after Theorem 1.
>
> In summary, we appreciate the reviewer to require further clarification on assumptions and realistic implications of our theorem. Our assumptions are pretty mild, commonly used in empirical Bayes, and are satisfied for the examples in Section 5. In addition, statistical literature including prior work in empirical Bayes chooses to keep dependency on these constants less explicit, and pays more attention to the sample complexity as we increase the sample size $n$.  We plan to carefully integrate more explanations and emphasize suitability of these assumptions to suitable locations in our final version, based on the discussion above. We also expect to extend our work further if we can leverage stronger properties.
> Please let us know if you have any further concerns and we are happy to address them in the next phase.
>
> ---
> [1] Koehler, Frederic, Alexander Heckett, and Andrej Risteski. "Statistical efficiency of score matching: The view from isoperimetry." arXiv preprint arXiv:2210.00726 (2022).

---

> > ### Comment · Reviewer_9mdT · 2025-08-04
> >
> > Thank you for the response. All of my concerns have been addressed and so I just raised my score.

---

> > > ### Author Response · Authors · 2025-08-05
> > >
> > > Thank you for your response and for raising the score. We sincerely appreciate your insightful comments and will be sure to carefully clarify our theoretical results in the revised manuscript. Thanks once again for your thoughtful feedback.

---

### Note · Authors · 2025-08-13

Dear AC and reviewers,

We sincerely thank you for your constructive reviews and respectful discussions.

We are encouraged that all reviewers find our framework novel, well-defined, rigorously guaranteed, and clearly presented. Reviewers also valued our framework for directly addressing over-shrinkage through multi-step denoising, as well as demonstrating clear empirical gains on both synthetic and real-world benchmarks.

We hope our responses have satisfied and addressed every reviewer's concerns.
We also confirm that the promised revisions can be successfully integrated into our final version, with main-text changes for critical clarity and appendices/limitations for extended discussion.
* We are strengthening the theory presentation by adding clarifying statements in Section 3 to better interpret the constant dependencies, verify practicality of the assumptions, and enhance the significance of our theory within the empirical Bayes theoretical framework.
* We append higher dimensional results in Section 5. Together with abundant existing results, we clarify and highlight our improved high-dimensional scalability from empirical observations. We then further discuss relevant theoretical conjecture in Limitation section.
* We agree that proving and mitigating the numerical approximation errors are important directions for future work. We thoroughly discuss these directions in the Limitation section and present supporting evidence in the supplemental.

The details of this revision plan are available in our responses to individual reviewers. We will also triple-check the manuscript to clear all typos and ensure notation clarity and consistency.

In summary, we have been refining our manuscript to directly address all reviewer concerns, with the goal of contributing a clear, rigorous, and useful advance to the community. We believe our work merits publication and discussion at the conference.

*Sincerely,*
The authors

---

### Decision · Program_Chairs · 2025-09-17

**Decision:**

Accept (poster)

**Comment:**

During the rebuttal period the authors addressed several concerns raised by reviewers, who all agree that the paper has important merits and deserves to be published. Hence I recommend an acceptance.